# TOPOLOGY OF ATTENTION DETECTS HALLUCINATIONS IN CODE LLMS

## ABSTRACT

While the AI-code assistant tools become widespread, automatic assessment of the correctness of the generated code becomes a significant challenge. Code LLMs are prone to hallucinations, which may lead to code that does not solve a required problem, or even to code with severe security vulnerabilities. In this paper, we propose a new approach to assessment of code correctness. Our solution is based on topological data analysis (TDA) of attention maps of code LLMs. We carry out experiments with two benchmarks – HumanEval, MBPP and 5 code LLMs: StarCoder2-7B, CodeLlama-7B, DeepSeek-Coder-6.7B, Qwen2.5-Coder-7B, Magicoder-S-DS-6.7B. The experimental results show that the proposed method is better than several baselines. Moreover, the trained classifiers are transferable between coding benchmarks.

## 1 INTRODUCTION

Large Language Models (LLMs) are now widespread and have great potential to transform natural language processing and artificial intelligence. As far as code generation is concerned, LLMs that are trained on large amounts of code are capable of generating human-level code for a plethora of simple problems and are expected to revolutionize software engineering. At the same time, code-generating LLMs are prone to hallucinations of various types. For example, syntactic and runtime errors prevent proper program execution, while logical errors lead to incorrect solution of the problem. In some cases, the generated code might contain security issues or robustness issues, such as a memory leak. While many definitions of hallucinations exist, in this paper we assume that code hallucination is a code which is not functionally correct, that is, does not pass functional tests. For a wide adoption of code LLMs, there is a high need for automatic assessment of code quality. Regarding the current state of technology, a significant amount of time is spent on debugging and automatic rewriting of generated code (Liang et al., 2024).

We hypothesize that code quality can be inferred before its execution from an internal state of LLM, in particular its attention maps. Previous studies have shown that transformer attention maps are useful for artificial text detection (Kushnareva et al., 2021), acceptability judgment (Cherniavskii et al., 2022) and speech classification (Tulchinskii et al., 2022).

Attention maps of LLMs are shown to capture semantically meaningful information and might be an illustration of the model's "thinking process". The research community actively studies approaches to mitigate hallucinations of LLMs by external knowledge bases (Peng et al., 2023) or to reduce them to some extent (Elaraby et al., 2023). It is highly desirable to evaluate the quality of the code before its execution and a run of tests since the code might contain security vulnerabilities.

The study of hallucinations in LLMs is intrinsically tied to generalization in NLP models. Both challenges stem from the way models learn, represent, and apply knowledge. Improving generalization through robust training, diverse data, and better uncertainty handling reduces hallucinations by ensuring that the models produce contextually appropriate and factually grounded output. In contrast, analyzing hallucinations provides an insight into generalization failures, guiding the development of more reliable NLP systems. This symbiotic relationship underscores the importance of addressing both issues holistically in AI research.

Out contributions are the following:

- We propose a new approach to detection of hallucinations in LLM-generated code based on analyzing a topology of attention maps;
- We carry out computational experiments with CodeLlama, StarCoder2, DeepSeek-Coder and Qwen2.5-Coder and two benchmarks – HumanEval and MBPP, and show that the proposed method outperforms baselines;
- We empirically show that the proposed hallucination classifier is transferable between code benchmarks.

## 2 RELATED WORK

Code generation via LLMs is a topic of active research. The popular projects are CodeLlama (Roziere et al., 2023), StarCoder2 (Lozhkov et al., 2024), DeepSeek-Coder (Guo et al., 2024), Qwen2.5-Coder (Hui et al., 2024), to name a few. Code LLMs differ by the data used for training, by their training and fine-tuning protocols, including RLHF, tokenizers, variants of attention mechanism, etc.

Several works studied attention maps in transformer-based LLMs. Clark et al. (2019) studied BERT's attention patterns: attending to delimiter tokens, specific positional offsets, or broadly attending over the whole sentence, with heads in the same layer often exhibiting similar behaviors. Clark et al. (2019) further showed that certain attention heads correspond well to the linguistic notions of syntax and coreference. Htut et al. (2019) found that for some universal dependency tree relation types, there exist heads that can recover the dependency type significantly better than baselines on parsed English text, suggesting that some self-attention heads act as a proxy for syntactic structure. Michel et al. (2019) showed that for downstream tasks, a large proportion of attention heads can be removed at test time without significantly affecting performance and that some layers can even be reduced to a single head.

The phenomenon of code hallucinations is studied and categorized in several papers. Tian et al. (2024) introduces a categorization of code hallucinations into four main types: mapping, naming, resource, and logic hallucinations, with each category further divided into different subcategories. Tian et al. (2024) proposed the CodeHalu dataset and studied the frequencies of different types of hallucinations in popular code LLMs. Liu et al. (2024) categorized hallucinations as intent conflicting, inconsistency, repetition, knowledge conflicting, dead code. Liu et al. (2024) released a HaluCode benchmark with code hallucinations labeled. Jiang et al. (2024) proposed Collu-Bench, the benchmark with localized code hallucinations. Jiang et al. (2024) found that code LLMs are less confident when hallucinating, since hallucinated tokens have a lower probability and hallucinated generation steps have a higher entropy. Tong & Zhang (2024) proposed to guide an LMM to work in the "slow thinking" regime to obtain a more accurate evaluation of generated code correctness.

In the broader context of NLP, several works introduced methods for preventing and detecting hallucinations. Peng et al. (2023) proposed to mitigate hallucination with an LLM-AUGMENTER, a system that allows the LLM to generate responses grounded in external knowledge, for example, stored in task-specific databases. Zhang et al. (2024b) proposed Self-Eval, a self-evaluation component, to prompt an LLM to validate the factuality of its own generated responses solely based on its internal knowledge. Feng et al. (2024) proposed two novel approaches for hallucination detection that are based on model collaboration, i.e., LLMs that investigate other LLMs for knowledge gaps, cooperatively or competitively. Zhang et al. (2024a) proposed to improve the truthfulness of LLMs by editing their internal representation during inference in the "truthful" space. Yehuda et al. (2024) introduced InterrogateLLM, a method that prompts the model multiple times to reconstruct the input query using the generated answer. Subsequently, InterrogateLLM quantifies the level of inconsistency between the original query and the reconstructed queries.

## 3 BACKGROUND

### 3.1 TRANSFORMER-BASED LLMS

All state-of-the art code LLMs are based on different variants of the transformer architecture (Vaswani, 2017).

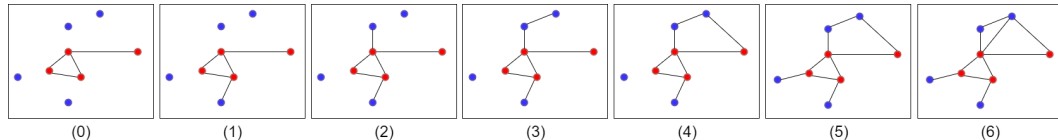

Figure 1: An example of MTD evaluation for a graph having two groups of vertices – red and blue. (0): initially, only edges connecting red vertices are present. (1)-(6): the rest of edges are added sequentially in an ascending order by their weights. While adding edges, connected components merge with each other. These moments are depicted by $H_0$ bars in Fig. 2. At moment (4) a cycle appears, at moment (6) this cycle disappears. These moments are depicted by the $H_1$ bar in Fig. 2.

A transformer architecture comprises $L$ layers of multi-head self-attention blocks, each of them having $H$ heads. Each attention head takes the matrix $X \in \mathbb{R}^{n \times d}$ as an input, and an output is $X^{out} = A(XW^V)$, where

$$A = \text{softmax}\left(\frac{(XW^Q)(XW^K)^T}{\sqrt{d}}\right),$$

and $W^Q, W^K, W^V \in \mathbb{R}^{d \times d}$ are projection matrices, and $A \in [0,1]^{n \times n}$ is an **attention map**. In the self-attention block, the attention

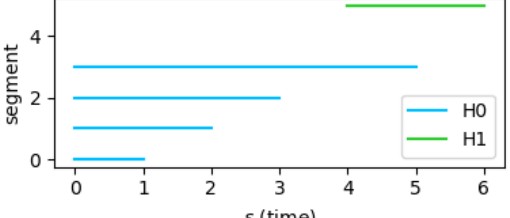

Figure 2: Cross-Barcode for a filtration from Fig. 1.

map shows how each token in the input sequence "interacts" with every other token in the same sequence. A token might attend more to other tokens that are contextually related. We interpret each element $a_{i,j}$ of an attention map as an "interaction force" between tokens $i$ and $j$.

## 3.2 REPRESENTING AN ATTENTION MAP BY A WEIGHTED GRAPH

While attention map is typically presented as a matrix, we treat it as a weighted graph. For $n$ tokens in a sequence, we consider a fully-connected weighted graph with $n$ vertices, where weights of edges are related to the "interaction force" between tokens (vertices). The natural idea is to leave only the most interacting tokens, that is, attending to each other higher than some threshold. However, the optimal threshold is not known in advance. Moreover, the topology of such a graph changes discontinuously with the change of a threshold (or weights). Topological Data Analysis (TDA) (Chazal & Michel, 2017) introduces a principled way to access the topology of such graphs for all thresholds simultaneously.

## 3.3 MANIFOLD TOPOLOGY DIVERGENCE

MTD (Manifold Topology Divergence) (Barannikov et al., 2021) is a tool of TDA that can be used to evaluate the "dissimilarity" between two sets of vertices in a weighted graph $\mathcal{G} = (V, E, W)$ or, in other words, to which degree one set of vertices is covered by another set.

Let a set of vertices $V = P \sqcup G$, be split into disjoint sets $P, G$. We consider a nested sequence of graphs $\mathcal{G}_0 \subset \ldots \subset \mathcal{G}_i \subset \mathcal{G}_{i+1} \subset \ldots \subset \mathcal{G}$ in the following way. $\mathcal{G}_0$ has all the vertices $P, G$ and all the edges that connect the vertices of $P$. The sequence $\mathcal{G}_i$ is obtained by adding the rest of the edges one by one in ascending order of their weights; see Figure 1. During this process, graph topology naturally changes: connected components are merged, cycles appear and disappear, etc. This process is rigorously described by the theory of *persistence barcodes* (Chazal & Michel, 2017). Each topological feature, such as connected component or cycle, has a "birth time" and a "death time", by a corresponding edge weight. The multi-set of these birth-death pairs (intervals) altogether is called a Cross-Barcode$_k$, see Figure 2. Here $k$ is an index of a *persistence homology*, each of them reflects a kind of topological feature: 0 - connected components, 1 - cycles, 2 - voids, etc. MTD$_k$ is an integral characteristic of a Cross-Barcode$_k$ and it is defined as a sum of birth-death intervals' lengths. The higher MTD$_k$ is, the greater is the "dissimilarity" between sets of tokens. Note that according to a definition, MTD$_k$ is not symmetric. Also, MTD$_k$, as a kind of persistence barcode, enjoys stability w.r.t. small perturbations of weights (Cohen-Steiner et al., 2005).

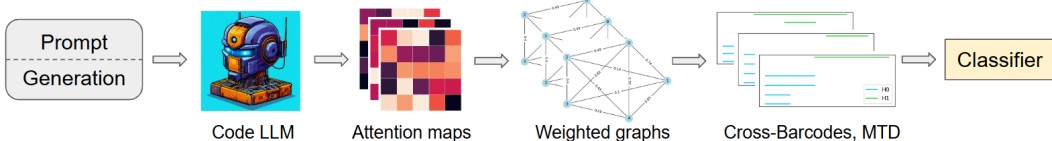

Figure 3: A pipeline of the proposed method for hallucination detection: (1) a prompt concatenated with a generated code is fed into a Code LLM. (2) Attention maps from the Code LLM are obtained. (3) Attention maps are transformed into fully-connected weighted graphs. (4) Cross-Barcodes and MTD features for weighted graphs are calculated. (5) On the top of the generated features a binary classifier of hallucinations is fitted.

Table 1: Characteristics of generated data: Pass@1, number of correct (#Pos.) and incorrect (#Neg.) solutions for each of the selected code LLMs.

| Model | HumanEval | | | MBPP | | |
|---|---|---|---|---|---|---|
| | Pass@1 | #Pos. | #Neg. | Pass@1 | #Pos. | #Neg. |
| StarCoder2-7B | 28.9 | 1186 | 2914 | 42.8 | 1071 | 1429 |
| CodeLlama-7B | 25.9 | 1064 | 3036 | 35.2 | 879 | 1621 |
| DeepSeek-Coder-6.7B | 40.3 | 1653 | 2447 | 52.6 | 1315 | 1185 |
| Qwen2.5-Coder-7B | 47.8 | 1961 | 2139 | 52.1 | 1302 | 1198 |
| Magicoder-S-DS-6.7B | 65.5 | 2689 | 1411 | 61.3 | 1533 | 967 |

## 4 METHODS

In the context of code generation, we naturally have two sets of tokens – a prompt and a generation. A common cause of hallucination is when the model's attention drifts away from the prompt[1]. Our topology-based features quantify this mismatch, yet the same signal helps to identify other error patterns as shown below (Section 5.6). As was pointed out in Section 3.2, attention matrices can be analyzed as weighted graphs. Specifically, for $n$ tokens in a sequence, we consider a fully-connected undirected weighted graph with $n$ vertices, where weights of edges are obtained from an attention map: $w_{i,j} = 1 - a_{i,j}$, for $i > j$ (we used decoder-only LLMs with causal attention). Then, Cross-Barcode and MTD for a weighted "attention graph" can be calculated[2]. To predict code hallucinations, we use the following set of features:

- $\text{MTD}_0(P, G)/|G|$, $\text{MTD}_0(G, P)/|P|$
- $\text{MTD}_1(P, G)/|G|$, $\text{MTD}_1(G, P)/|P|$
- $\sum_{i \in P} a_{i,i}/|P|$, $\sum_{i \in G} a_{i,i}/|G|$

Here, all features are normalized by the size of the set of corresponding vertices for better transferability. In addition, sums of diagonal values of the attention matrices that are not directly present in edge weights are included. These features are calculated for every layer and head of a code LLM. At the top of the proposed topological features, we applied XGBoost (Chen & Guestrin, 2016) for a classification[3]. The high-level pipeline of the proposed method is shown in Figure 3.

MTD, when applied to attention graphs, measures the strength of connectivity between tokens of the generation and tokens of the prompt. Low values of MTD mean the high connectivity 1) between tokens of the generation and the prompt 2) between tokens inside the generation. MTD scores of some heads have a significant discriminative power and are shown in Fig. 4. Lack of generation-prompt attention means that representations of tokens from generation does not depend on prompt. That is, LLM drifts away from the prompt during generation and hallucinates.

---

[1]The definition of hallucination is discussed in the Appendix G.

[2]Appendix D contains examples of Cross-Barcodes and corresponding attention maps.

[3]Appendix E contains an ablation study of a classifier model.

# 5 EXPERIMENTS

## 5.1 GENERATION OF DATASETS

To assess the efficacy of the proposed method for hallucination prediction, we carried out a set of computational experiments. In the main experiments, we use the following popular code LLMs: StarCoder2-7B (Lozhkov et al., 2024), CodeLlama-7B (Roziere et al., 2023), DeepSeek-Coder-6.7B (Guo et al., 2024), Qwen2.5-Coder-7B (Hui et al., 2024), Magicoder-S-DS-6.7B (Wei et al., 2024). We adapted two public benchmarks for evaluation of code generation: HumanEval (Chen et al., 2021) and MBPP (Austin et al., 2021) [4]. In order to account for various possible code generations, for each of the coding problems, several solutions were generated by each of the selected code LLMs: we obtained 25 generations per task for HumanEval and 5 generations per task for MBPP unless otherwise specified. To address the quality of the proposed approach in different LLM prompting regimes, we used a 0-shot prompt for the HumanEval dataset and a 1-shot prompt for the MBPP dataset. To enable diversity of the generated solutions, a temperature sampling was done (see Appendix L for evaluation with greedy decoding.) Thus, we obtain 4100 samples for HumanEval and 2500 samples for MBPP for each code LLM (see Appendix A for further details). Table 1 presents a summary of code solutions generated. The correctness of the code is evaluated via functional tests provided together with the coding benchmarks. Functional tests check that the function called with certain arguments has the corresponding output (examples are shown in Figures 6, 7 in the Appendix A). Incorrect code is considered a "hallucination"; prediction of code's correctness is a binary classification problem. The pass@1 metric is slightly lower that reported in original papers, mostly because we have used sampling with non-zero temperature instead of greedy search. Before moving further, note that there is a strong negative dependency between prompt and generation lengths and code quality; see Figure 8, 9. The longer the prompt (i.e. task description) and generation (i.e. task solution) are, the lower is the probability of code's correctness. This dependency is more pronounced for HumanEval than for MBPP, because MBPP employed more complicated 1-shot prompts. These attributes are natural baselines for hallucination's prediction.

## 5.2 ANALYZING METHOD'S CLASSIFICATION QUALITY

Using the generated data, we estimate the classification quality of the proposed approach. We applied 5-fold stratified group cross-validation where different solutions of the same coding problem belonged to the same group. In this way, training and testing were performed always at non-overlapping coding problems (prompts). The reported results are the mean and standard deviation estimated over the 5 folds.

As baselines for comparison, we used the XGBoost classifier trained on simple features: tokenized prompt length, tokenized generation length, and mean log. probability of generated tokens Chen et al. (2021). We provide a comparison with Pylint[5], a static code analysis tool for Python. In addition, we trained a linear classification head on top of a frozen CodeT5-base encoder Wang et al. (2021). Furthermore, we have adapted Self-Eval Zhang et al. (2024b) and Interrogate-LLM Yehuda et al. (2024) to detect hallucinations in LLM-generated code and also utilized them as baselines. Finally, we utilize a combination of all features, i.e. tokenized prompt length, tokenized generation length, mean log. probability, and the proposed attention features, to train a classifier (we refer to it 'All Feat.' for brevity). Training details are provided in Appendix B. Table 2 presents the main results. Table 12 in the Appendix presents the additional results for the $32 - 34B$ models.

In the majority of cases, the proposed classifier based on the features of the attention maps performed significantly better than the baselines and demonstrated stable results for all models and datasets as measured by the ROC-AUC score. Further analysis revealed that some features made a significant contribution to classification quality; see Figure 4. The use of additional features can both decrease and increase overall performance. Thus, we believe that the proposed attention features are strong enough to capture the most important information for code hallucination detection.

---

[4] Licenses of pretrained models and benchmarks permit use for research purposes.
[5] https://www.pylint.org/

Table 2: Quality of code hallucination detection for HumanEval and MBPP datasets. **Bold** and underline denote the first and second best results.

| Model | HumanEval | | MBPP | |
|---|---|---|---|---|
| | ROC-AUC | F1-Score | ROC-AUC | F1-Score |
| StarCoder2-7B | | | | |
| Prompt Len. | $54.5 \pm 6.6$ | $24.6 \pm 11.2$ | $51.2 \pm 2.3$ | $40.0 \pm 3.8$ |
| Gen. Len. | $57.7 \pm 5.6$ | $13.5 \pm 4.8$ | $57.7 \pm 0.9$ | $45.4 \pm 3.5$ |
| Mean Log. Prob. | $70.9 \pm 1.3$ | $32.4 \pm 4.8$ | $62.0 \pm 2.0$ | $47.5 \pm 3.4$ |
| Pylint | $58.9 \pm 1.0$ | $49.6 \pm 4.7$ | $53.0 \pm 0.6$ | $61.4 \pm 1.0$ |
| CodeT5-base ft. | $70.1 \pm 7.1$ | $33.3 \pm 10.1$ | $58.5 \pm 3.5$ | $43.3 \pm 9.0$ |
| Self-Eval | $50.0 \pm 2.9$ | $0.0 \pm 0.0$ | $58.9 \pm 2.6$ | $0.0 \pm 0.0$ |
| Interrogate-LLM | $63.9 \pm 2.1$ | $48.1 \pm 4.2$ | $58.6 \pm 2.6$ | $60.9 \pm 1.3$ |
| Attn. Feat. (ours) | $\underline{82.9 \pm 2.7}$ | $\underline{54.2 \pm 6.9}$ | $\mathbf{81.9 \pm 2.4}$ | $\underline{68.4 \pm 5.3}$ |
| All Feat. | $\mathbf{83.7 \pm 4.1}$ | $\mathbf{57.5 \pm 7.9}$ | $\underline{81.8 \pm 2.1}$ | $\mathbf{68.7 \pm 5.1}$ |
| CodeLlama-7B | | | | |
| Prompt Len. | $61.6 \pm 4.4$ | $25.7 \pm 15.0$ | $59.1 \pm 4.2$ | $35.4 \pm 2.9$ |
| Gen. Len. | $60.1 \pm 5.3$ | $10.6 \pm 7.0$ | $60.8 \pm 2.5$ | $24.2 \pm 5.3$ |
| Mean Log. Prob. | $64.1 \pm 2.0$ | $25.4 \pm 6.2$ | $61.0 \pm 3.7$ | $27.2 \pm 1.5$ |
| Pylint | $55.1 \pm 0.3$ | $43.7 \pm 4.8$ | $53.1 \pm 0.5$ | $53.5 \pm 1.1$ |
| CodeT5-base ft. | $74.5 \pm 6.3$ | $43.6 \pm 13.2$ | $61.7 \pm 3.0$ | $19.1 \pm 7.1$ |
| Self-Eval | $49.7 \pm 1.7$ | $39.6 \pm 4.4$ | $50.0 \pm 0.0$ | $0.0 \pm 0.0$ |
| Interrogate-LLM | $67.9 \pm 2.5$ | $44.5 \pm 4.2$ | $57.6 \pm 2.4$ | $52.5 \pm 1.6$ |
| Attn. Feat. (ours) | $\underline{85.6 \pm 3.9}$ | $\underline{56.4 \pm 7.2}$ | $\underline{83.4 \pm 3.3}$ | $\underline{64.0 \pm 4.4}$ |
| All Feat. | $\mathbf{85.7 \pm 3.8}$ | $\mathbf{57.9 \pm 9.7}$ | $\mathbf{83.8 \pm 2.0}$ | $\mathbf{65.2 \pm 4.3}$ |
| DeepSeek-Coder-6.7B | | | | |
| Prompt Len. | $56.2 \pm 4.6$ | $44.4 \pm 4.3$ | $52.5 \pm 2.5$ | $56.4 \pm 3.6$ |
| Gen. Len. | $57.9 \pm 2.4$ | $34.4 \pm 4.9$ | $54.6 \pm 1.9$ | $59.4 \pm 1.3$ |
| Mean Log. Prob. | $69.8 \pm 2.5$ | $51.1 \pm 3.4$ | $61.0 \pm 1.9$ | $62.3 \pm 1.6$ |
| Pylint | $54.6 \pm 0.8$ | $59.7 \pm 3.1$ | $52.8 \pm 0.4$ | $70.1 \pm 1.6$ |
| CodeT5-base ft. | $69.1 \pm 4.2$ | $52.6 \pm 6.5$ | $55.7 \pm 3.0$ | $64.8 \pm 2.7$ |
| Self-Eval | $56.7 \pm 2.5$ | $0.0 \pm 0.0$ | $51.1 \pm 0.6$ | $69.2 \pm 1.6$ |
| Interrogate-LLM | $71.8 \pm 2.4$ | $62.9 \pm 2.7$ | $60.1 \pm 5.1$ | $69.0 \pm 1.8$ |
| Attn. Feat. (ours) | $\mathbf{85.6 \pm 2.8}$ | $\underline{68.9 \pm 5.5}$ | $\underline{82.6 \pm 1.9}$ | $\underline{76.5 \pm 2.7}$ |
| All Feat. | $\underline{85.4 \pm 2.7}$ | $\mathbf{69.3 \pm 5.3}$ | $\mathbf{83.1 \pm 1.2}$ | $\mathbf{77.0 \pm 1.3}$ |
| Qwen2.5-Coder-7B | | | | |
| Prompt Len. | $54.3 \pm 8.7$ | $51.0 \pm 5.7$ | $51.8 \pm 3.6$ | $56.2 \pm 4.4$ |
| Gen. Len. | $57.6 \pm 3.6$ | $48.9 \pm 5.1$ | $55.6 \pm 2.1$ | $59.7 \pm 4.8$ |
| Mean Log. Prob. | $63.1 \pm 2.4$ | $55.6 \pm 5.5$ | $61.5 \pm 1.3$ | $60.4 \pm 1.8$ |
| Pylint | $64.1 \pm 2.2$ | $\underline{71.4 \pm 6.3}$ | $62.0 \pm 2.5$ | $74.1 \pm 0.8$ |
| CodeT5-base ft. | $65.9 \pm 3.7$ | $58.2 \pm 4.5$ | $56.0 \pm 1.3$ | $65.2 \pm 2.0$ |
| Self-Eval | $73.8 \pm 1.6$ | $\mathbf{75.6 \pm 3.8}$ | $66.6 \pm 2.3$ | $\underline{76.0 \pm 1.3}$ |
| Interrogate-LLM | $60.1 \pm 3.7$ | $65.7 \pm 5.1$ | $55.0 \pm 1.8$ | $68.3 \pm 2.1$ |
| Attn. Feat. (ours) | $\underline{81.7 \pm 2.8}$ | $70.2 \pm 4.2$ | $\underline{82.2 \pm 2.2}$ | $75.4 \pm 1.7$ |
| All Feat. | $\mathbf{81.8 \pm 2.3}$ | $69.6 \pm 3.0$ | $\mathbf{82.3 \pm 2.0}$ | $\mathbf{76.7 \pm 1.7}$ |
| Magicoder-S-DS-6.7B | | | | |
| Prompt Len. | $57.3 \pm 5.4$ | $70.4 \pm 7.0$ | $54.0 \pm 2.6$ | $69.2 \pm 3.0$ |
| Gen. Len. | $52.5 \pm 2.1$ | $76.3 \pm 2.6$ | $52.3 \pm 2.1$ | $71.4 \pm 2.4$ |
| Mean Log. Prob. | $71.0 \pm 5.3$ | $78.4 \pm 2.9$ | $60.5 \pm 3.7$ | $71.4 \pm 2.1$ |
| Pylint | $52.7 \pm 1.3$ | $80.0 \pm 3.3$ | $52.0 \pm 0.8$ | $\underline{76.7 \pm 2.4}$ |
| CodeT5-base ft. | $64.7 \pm 2.7$ | $77.5 \pm 2.7$ | $61.0 \pm 3.7$ | $74.8 \pm 1.4$ |
| Self-Eval | $44.3 \pm 3.2$ | $79.1 \pm 3.1$ | $49.2 \pm 1.4$ | $75.5 \pm 2.4$ |
| Interrogate-LLM | $65.3 \pm 2.1$ | $79.6 \pm 3.1$ | $59.0 \pm 7.1$ | $76.1 \pm 2.1$ |
| Attn. Feat. (ours) | $\mathbf{82.3 \pm 4.9}$ | $\mathbf{80.7 \pm 3.6}$ | $\mathbf{81.5 \pm 1.6}$ | $\mathbf{80.6 \pm 2.0}$ |
| All Feat. | $\underline{81.8 \pm 3.3}$ | $\mathbf{80.7 \pm 3.0}$ | $\underline{81.2 \pm 2.0}$ | $\mathbf{80.6 \pm 2.2}$ |

Moreover, the classifier detects various failure modes: SyntaxError, ZeroDivisionError, NameError, etc., across six categories (Section 5.6, Table 5), confirming that the approach generalizes beyond the prompt-insufficient attention failure mode.

Table 3: pass@1 scores for variants of ranking of code generations.

| Model | HumanEval | | MBPP | |
|---|---|---|---|---|
| | Random | Clf. Prob. | Random | Clf. Prob. |
| StarCoder2-7B | $28.6 \pm 5.5$ | $\mathbf{43.3 \pm 9.0}$ | $43.0 \pm 3.6$ | $\mathbf{49.6 \pm 4.6}$ |
| CodeLlama-7B | $26.0 \pm 5.1$ | $\mathbf{39.7 \pm 7.2}$ | $35.2 \pm 3.3$ | $\mathbf{43.6 \pm 3.4}$ |
| DeepSeek-Coder-6.7B | $39.1 \pm 4.9$ | $\mathbf{56.7 \pm 7.4}$ | $53.0 \pm 2.5$ | $\mathbf{61.4 \pm 2.3}$ |
| Qwen2.5-Coder-7B | $51.8 \pm 8.0$ | $\mathbf{64.0 \pm 7.3}$ | $52.6 \pm 3.6$ | $\mathbf{62.0 \pm 2.4}$ |
| Magicoder-S-DS-6.7B | $72.5 \pm 10.0$ | $\mathbf{74.3 \pm 6.1}$ | $61.4 \pm 3.4$ | $\mathbf{64.8 \pm 2.1}$ |

Furthermore, we evaluated the proposed approach on Java (high resource), Go (medium resource), Rust (low resource), and Lua (niche) programming languages of the HumanEval subdivision of the MultiPL-E dataset Cassano et al. (2023). The proposed Attn. Feat. classifier can detect hallucinations even in low resource and niche languages, and there are several separate features that perform robust across the languages; see Tables 17, 19, 20. Furthermore, Table 16 demonstrates that the proposed Attn. Feat. classifier has consistent quality when the linguistic complexity of the prompt is increased. Finally, for additional comparison with zero-shot classification via larger LLMs, see Appendix J.

## 5.3 ANALYZING METHOD'S RANKING QUALITY

Next, we assess the usefulness of the proposed code hallucination classifier for ranking of code generations. For each problem, all generations were ranked according to the predicted probability of correctness and one with the highest probability was selected. A baseline was random selection of a code generation. The use of a classifier always leads to a significantly higher pass@1 score; see Table 3. This result justifies that the proposed Att. Feat. classifiers can distinguish different levels of code correctness among several candidate solutions to the same problem. Even for more recent Qwen2.5-Coder-7B and Magicoder-S-DS-6.7B, pass@1 can still be further improved with the use of Attn. Feat. classifier.

## 5.4 METHOD'S TRANSFERABILITY BETWEEN BENCHMARKS

We further study the transferability of the classifiers based on topological features. In this setting, hallucination classifiers for a fixed code LLM were trained on data for one benchmark (HumanEval, MBPP) and evaluated on another, then repeated vise versa. Table 4 shows the results: the proposed classifiers are transferable, although performance is lower when training and testing is done on the same benchmark. The proposed attention features achieve better transferability in 70% of cases, as measured by ROC-AUC for both the HE $\rightarrow$ MBPP and MBPP $\rightarrow$ HE transfer. We relate the changes in the classifiers' performance compared to results from Section 5.2 to differences in the prompt structure: we use 0-shot prompt for HumanEval and 1-shot prompt for MBPP. Next, it is possible to further improve transferability by combining the proposed attention features with the mean log. probability of a generation. The combined classifier outperforms the Mean. Log. Prob. baseline in 80% of cases. This indicates that these two approaches cover different aspects, and the proposed attention features are indeed useful.

Next, we study cross-model transferability of classifiers. A classifier trained on DeepSeek-Coder-6.7B is transferable to Magicoder-S-DS-6.7B and vice versa. But it is not the case for CodeLlama-7B. The reason is that Magicoder-S-DS-6.7B is a fine-tuned DeepSeek-Coder-6.7B and a correspondence between attention heads persists, see Appendix M.

## 5.5 ANALYZING FEATURE IMPORTANCE

In its base setup, the proposed approach requires computation of attention features from attention maps for all layers and heads. However, we observed that the trained XGBoost classifier experienced a natural sparsity with only about 25% meaningful features, as measured by the feature importance attributed by the classifier. To further explore the importance and selection of features, we followed

Table 4: Transferability of code hallucination detectors. Each classifier was trained on HumanEval (MBPP) dataset and tested on MBPP (HumanEval) dataset.

| Model | HumanEval → MBPP | | MBPP → HumanEval | |
|---|---|---|---|---|
| | ROC-AUC | F1-Score | ROC-AUC | F1-Score |
| *StarCoder2-7B* | | | | |
| Prompt Len. | 48.6 | 0.0 | 52.1 | 45.0 |
| Gen. Len. | 56.0 | 14.6 | 52.4 | 38.3 |
| Mean Log. Prob. | 63.7 | **36.2** | **71.8** | **45.4** |
| CodeT5-base ft. | 53.7 | 0.0 | 59.1 | 0.0 |
| Attn. Features | 67.5 | 0.14 | 67.2 | 25.5 |
| Attn. Feat. (ours) + Mean. Log. Prob | **71.0** | 24.0 | 61.7 | 15.7 |
| *CodeLlama-7B* | | | | |
| Prompt Len. | 51.7 | 0.0 | 53.4 | **42.9** |
| Gen. Len. | 61.5 | 4.2 | 50.0 | 41.0 |
| Mean Log. Prob. | 57.7 | **15.2** | 65.0 | 34.9 |
| CodeT5-base ft. | 54.9 | 0.0 | 62.4 | 0.0 |
| Attn. Features | 69.5 | 0.2 | **80.3** | 34.1 |
| Attn. Feat. (ours) + Mean. Log. Prob. | **72.3** | 15.1 | 79.0 | 34.9 |
| *DeepSeek-Coder-6.7B* | | | | |
| Prompt Len. | 48.0 | 15.6 | 52.2 | 58.2 |
| Gen. Len. | 55.3 | 41.4 | 54.0 | 51.3 |
| Mean Log. Prob. | 62.5 | 56.3 | 69.1 | **58.3** |
| CodeT5-base ft. | 53.4 | 0.0 | 55.9 | 57.4 |
| Attn. Features | 67.7 | 70.4 | 72.4 | 20.4 |
| Attn. Feat. (ours) + Mean. Log. Prob. | **68.0** | **71.0** | **72.5** | 22.6 |
| *Qwen2.5-Coder-7B* | | | | |
| Prompt Len. | 49.9 | 34.1 | 51.1 | 64.1 |
| Gen. Len. | 51.6 | 46.1 | 54.5 | 54.3 |
| Mean Log. Prob. | 60.3 | 60.4 | 64.7 | 60.8 |
| CodeT5-base ft. | 49.1 | 52.3 | 51.6 | **65.6** |
| Attn. Features | **70.6** | **63.3** | 64.2 | 54.3 |
| Attn. Feat. (ours) + Mean. Log. Prob. | 70.0 | 55.7 | **65.4** | 56.1 |
| *Magicoder-S-DS-6.7B* | | | | |
| Prompt Len. | 48.1 | 56.3 | 54.5 | **79.5** |
| Gen. Len. | 54.8 | 75.2 | 56.1 | 74.6 |
| Mean Log. Prob. | 63.7 | 74.9 | **69.8** | 76.5 |
| CodeT5-base ft. | 49.3 | 76.0 | 45.9 | 79.2 |
| Attn. Features | **73.5** | 78.4 | 56.9 | 37.6 |
| Attn. Feat. (ours) + Mean. Log. Prob. | 71.8 | **79.1** | 63.7 | 45.8 |

the two-stage pipeline. First, for a given sparsity level, we selected the most important features as measured by the feature importance attributed by the classifier trained on all attention features simultaneously. Second, we trained a new XGBoost classifier on the selected set of attention features. As indicated by Figures 5, 10, the proposed feature selection procedure could retain only 5% of all attention features without a significant loss of classification quality, highlighting that only a limited number of all attention heads are relevant for hallucination detection.

We carry out additional experiments benchmarking different programming languages (Python, Go, Rust, Java) and find that topological features of some heads exhibit a high predictive performance consistently across all programming languages; see the Appendix K.

## 5.6 A FINE-GRAINED CLASSIFICATION OF ERROR TYPES

We carried out additional experiments to study the ability of the proposed approach to detect specific types of hallucinations. An exception in Python can be considered as a hallucination type. Here are the common exceptions from the HumanEval and MBPP benchmarks: *AssertionError, AttributeEr-*

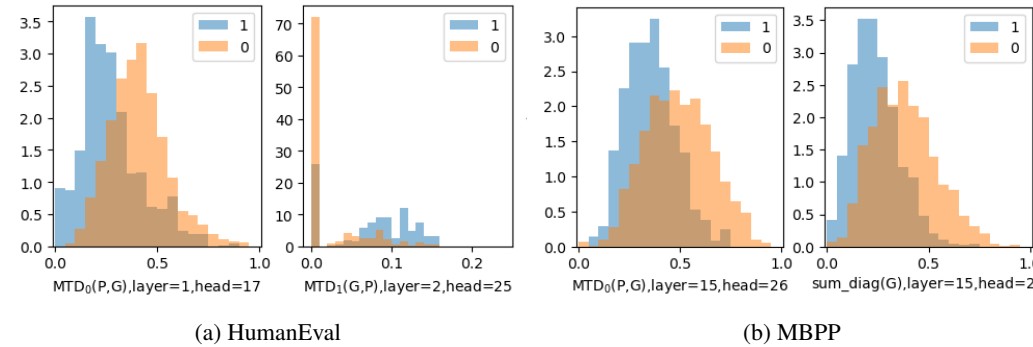

(a) HumanEval                    (b) MBPP

Figure 4: Distribution of classes (0-code is not correct, hallucination, 1-code is correct) vs. features from attention maps. Some of the most discriminative features are presented. Features are normalized with MinMaxScaler. Features come from CodeLlama-7B.

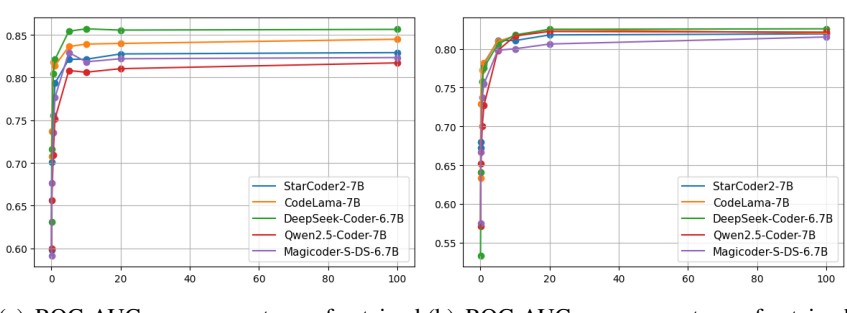

(a) ROC-AUC vs. percentage of retained features, HumanEval.

(b) ROC-AUC vs. percentage of retained features, MBPP.

Figure 5: Analysis of feature importance of the proposed method, ROC-AUC.

*ror, IndentationError, IndexError, ModuleNotFoundError, NameError, RecursionError, SyntaxError, TypeError, UnboundLocalError, ValueError, ZeroDivisionError, timed out*

Therefore, we did multi-classification instead of binary classification in XGBoost. Table 5 shows the results. This result demonstrates that the proposed attention features are capable of identifying the majority of errors of different types, consistently across the models. Here is a breakdown of detection accuracy of particular types of errors for CodeLlama-7B: AssertionError: 82.0%, IndexError: 97.8%, NameError: 75.7%, RecursionError: 100%, SyntaxError: 93.3%, TypeError: 80.6%, ValueError: 87.5%, ModuleNotFoundError, ZeroDivisionError, UnboundLocalError, IndentationError, AttributeError, timed out : 80%. Some error types were grouped because of a very low frequency.

## 5.7 ABLATION STUDY

The proposed approach is based on the two types of attention features: the diagonal elements of attention maps corresponding to a prompt and a generation, and topological features (MTD) computed for the corresponding "attention graph" (see Section 4 for details). In this Section, we provide an ablation study to estimate the contribution of each type of attention features. For this purpose, we trained the XGBoost classifier using 1) only MTD features, 2) only diagonal attention values, and 3) both types of features (our initial setup). As demonstrated in Table 9 in the Appendix, DeepSeek-Coder-6.7B and Qwen2.5-Coder-7B achieved the best performance when both types of attention features were used for HumanEval and MBPP datasets. In contrast, the best performance of StarCoder2-7B and Magicoder-S-DS-6.7B was achieved with different sets of attention features depending on the dataset and metric choices. In order to account for various information available via attention maps, we propose using both types of features as the most universal choice. Never-

Table 5: Performance of multi-classification of error types.

| Model | Accuracy | F1-Score |
|---|---|---|
| StarCoder2-7B | $0.7 \pm 0.02$ | $0.68 \pm 0.02$ |
| CodeLlama-7B | $0.66 \pm 0.02$ | $0.62 \pm 0.02$ |
| DeepSeek-Coder-6.7B | $0.7 \pm 0.03$ | $0.68 \pm 0.04$ |
| Qwen2.5-Coder-7B | $0.64 \pm 0.02$ | $0.6 \pm 0.02$ |
| Magicoder-S-DS-6.7B | $0.73 \pm 0.02$ | $0.71 \pm 0.02$ |

theless, we note that for some code LLM one certain type of attention features may result in better performance than combination of both types[6].

## 6 CONCLUSIONS

In this paper, we propose a new hallucination detection approach for code-generating LLMs. Our approach is based on the introspection of an LLM: we get attention maps for a prompt and generation and study their topology after transforming to weighed graphs. The proposed topological features of these graphs have been empirically shown to be relevant for the detection of code hallucinations. A classifier built on top of these features outperformed several baselines. These classifiers are transferable across the coding benchmarks. The natural extension of our research is the detection of specific places of code with bugs, and we leave it for further research. We believe that our work may lead to a wider application of code LLMs by making them more reliable. In a wider context, our work contributes to the study of interpretation and generalization in NLP models, since hallucinations and generalization ability are intrinsically tied.

---

[6]Additional ablation study in Appendix I shows that for small models (QwenCoder-1.5b, QwenCoder-3b), and transfer learning setting, the differences in contribution of diagonal and MTD features is more pronounced.

## 7 REPRODUCIBILITY STATEMENT

The source code to reproduce the results presented is provided in the supplementary material. Hyperparameters are either disclosed in the main text and Appendices A, B, or were equal to default values in the code.

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

```
from typing import List

def has_close_elements(numbers: List[float], threshold: float) -> bool:
    """ Check if in given list of numbers, are any two numbers closer to each other than
    given threshold.
    >>> has_close_elements([1.0, 2.0, 3.0], 0.5)
    False
    >>> has_close_elements([1.0, 2.8, 3.0, 4.0, 5.0, 2.0], 0.3)
    True
    """
```
*problem description*

```
    for i in range(len(numbers) - 1):
        for j in range(i+1, len(numbers)):
            if abs(numbers[i] - numbers[j]) <= threshold:
                return True
    return False
```
*generation*

Figure 6: Example of prompt (problem description) and model generation for the HumanEval dataset.

```
You are an expert Python programmer, and here is your task: Write a function to find the
similar elements from the given two tuple lists. Your code should pass these tests:

assert similar_elements((3, 4, 5, 6),(5, 7, 4, 10)) == (4, 5)
assert similar_elements((1, 2, 3, 4),(5, 4, 3, 7)) == (3, 4)
assert similar_elements((11, 12, 14, 13),(17, 15, 14, 13)) == (13, 14)
[BEGIN]
def similar_elements(test_tup1, test_tup2):
  res = tuple(set(test_tup1) & set(test_tup2))
  return (res)
[DONE]
```
*one-shot*

```
You are an expert Python programmer, and here is your task: Write a python function to
remove first and last occurrence of a given character from the string. Your code should
pass these tests:

assert remove_Occ("hello","l") == "heo"
assert remove_Occ("abcda","a") == "bcd"
assert remove_Occ("PHP","P") == "H"
[BEGIN]
```
*problem description*

```
def remove_Occ(s,c):
  return s.replace(c,'',s.count(c)-1)
[DONE]
```
*generation*

Figure 7: Example of prompt (one-shot example and problem description) and model generation for the MBPP dataset.

# A  DETAILS ON GENERATION PROCEDURE

We generate solutions for the coding problems with a temperature of $0.8$. For the HumanEval dataset, the maximum length of the model output (i.e., input prompt + generation) was limited to $512$ tokens. For the MBPP dataset, the maximum number of new tokens to generate was set to $256$. Figures 6, 7 provide examples of prompts and generations for HumanEval and MBPP datasets. We followed the guidelines[7] to post process the model output and extract the valid problem solution. To compute the attention features according to the method proposed in Section 4, we used the attention submatrix corresponding to the input prompt and the valid solution to the problem. For computational experiments, we used NVIDIA TITAN RTX.

---

[7]https://github.com/bigcode-project/bigcode-evaluation-harness

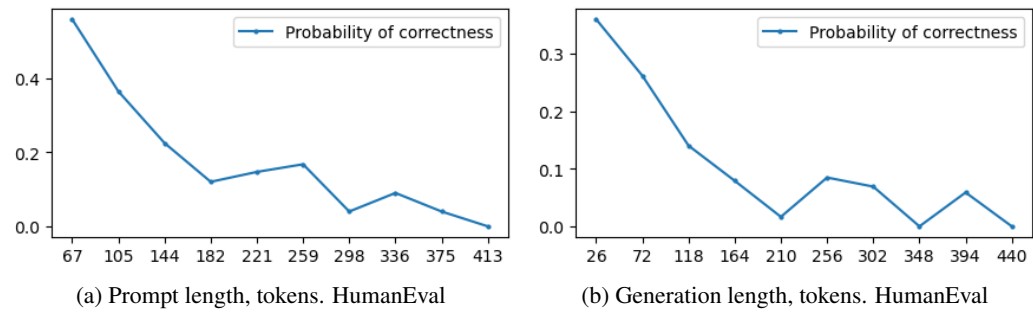

(a) Prompt length, tokens. HumanEval    (b) Generation length, tokens. HumanEval

Figure 8: The individual conditional expectations for prompt and generation lengths, CodeLlama-7B.

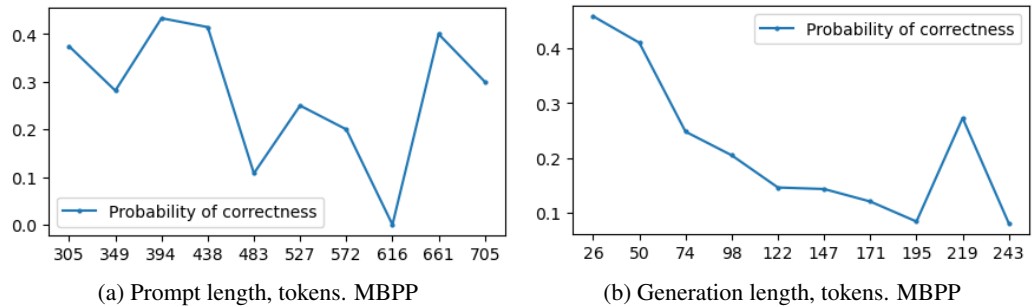

(a) Prompt length, tokens. MBPP    (b) Generation length, tokens. MBPP

Figure 9: The individual conditional expectations for prompt and generation lengths, CodeLlama-7B.

## B    DETAILS ON TRAINING PROCEDURE

For the code hallucination detectors, based on the XGBoost classifier training, we utilized the XGB-Classifier with an approximation tree method "hist" from the XGBoost library[8]. For the code hallucination detector based on CodeT5-base embeddings, we used the pre-trained frozen CodeT5-base encoder with a trainable classification head consisting of 2 linear layers with hidden dimensionality 768. The classification head was trained for 100 epochs with batch size 32 and learning rate $3e-5$.

Self-Eval (Zhang et al., 2024b) is a way to evaluate the responses of an LLM using its internal knowledge. Self-Eval extracts a list of atomic claims from the responses and then prompts an LLM itself to validate the factuality of the claims. Self-Eval is not directly applicable to Code LLMs, since there are no "facts" in the code. However, we applied the core idea of Self-Eval by prompting Code LLMs to evaluate the functional correctness of a generated code. In addition, we have adapted Interrogate-LLM (Yehuda et al., 2024) to detect hallucinations in LLM-generated code. As an embedding model, we used the CodeT5+ 110M Embedding model, $K = 5$ and a fixed temperature.

## C    METRICS USED FOR EVALUATION

In order to account for possible class imbalance, we use ROC-AUC and F1-Score to evaluate the code hallucination detectors. We briefly introduce them in the following.

The ROC curve demonstrates the quality of a binary classifier for all possible classification thresholds. The X-axis corresponds to the False Positive Rate (FPR) and the Y-axis corresponds to the True Positive Rate (TPR) which can be defined as follows: $FPR = \frac{FP}{FP+TN}, TPR = \frac{TP}{FP+FN}$, where TP – true positive samples, FP – false positive samples, TN – true negative samples, FN –

---
[8]https://xgboost.readthedocs.io/en/latest/index.html

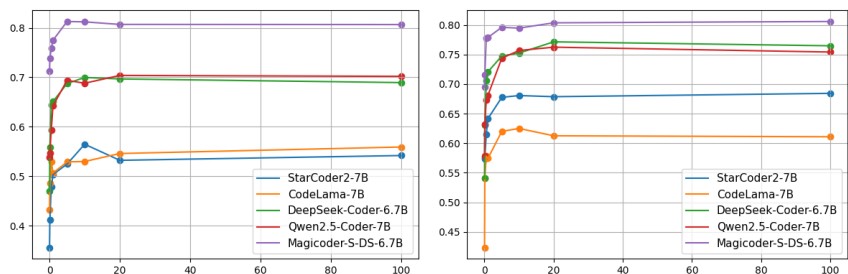

(a) F1 vs. percentage of retained features, HumanEval.  (b) F1 vs. percentage of retained features, MBPP.

Figure 10: Analysis of feature importance of the proposed method, F1-score

false negative samples. ROC-AUC is defined as the area under the ROC curve. ROC-AUC of a random model is equal to 0.5, ROC-AUC of a perfect model is 1.

The F1-score is a harmonic mean of Precision and Recall:

$$F_1 = \frac{2}{\frac{1}{Precision} + \frac{1}{Recall}}.$$

To study the ranking ability of the hallucinations detector, we used the pass@1 metric. pass@1 is a proportion of coding problems from a benchmark for which a Code LLM generated the correct solution passing all the tests, with the restriction that only one solution (among 25 generations for HumanEval and 5 for MBPP) is executed.

## D    EXAMPLES OF CROSS-BARCODES

Figure 11 shows examples of Cross-Barcode$_0$ for a fixed attention head. The Cross-Barcode$_1$(P, G) is empty for these attention maps. Correct generations (a), (b) tend to have more and more H$_0$ bars than not-correct ones (c), (d).

Figure 12 shows examples of Cross-Barcode$_0$, Cross-Barcode$_1$ for a fixed attention head. Correct generations (a), (b) tend to have more and longer H$_1$ bars than not-correct ones (c), (d).

Attention maps for the corresponding head are shown in Figures 13, 14.

## E    EXPERIMENTS ON THE CLASSIFIER MODEL SELECTION

We carried out additional experiments with the feed-forward network (MLP), logistic regression, and support vector classifier (SVC) instead of XGBoost as a classifier for hallucination detection (the rightmost block in Figure 3). We used MLP with two hidden layers of size 256 and ReLU activations. This configuration was selected after moderate optimization of an architecture. For logistic regression and SVC, we tuned the value of regularization strength. Table 6 presents the results. MLP tends to have a lower ROC-AUC, but sometimes has a higher F1-score. While the F1-score is a threshold-dependent metric, the ROC-AUC integrates over all thresholds. In an unbalanced setting, ROC-AUC is often more stable, thus a classifier may have a higher ROC-AUC even when its F1-score is lower. XGBoost offers (a) strong average performance across all code LLMs, (b) negligible training cost ($\approx$ 30 s per fold), and (c) no hyper-parameter tuning in our setting. XGBoost guaranties low computational overhead while providing a single, robust baseline for subsequent work.

## F    A NOTE ON APPLICABILITY OF GNNS TO ATTENTION MAPS

To train a GNN-based approach on graphs with edge weights obtained from attention matrices, these attention matrices need to be stored. We can estimate the approximate memory footprint to store

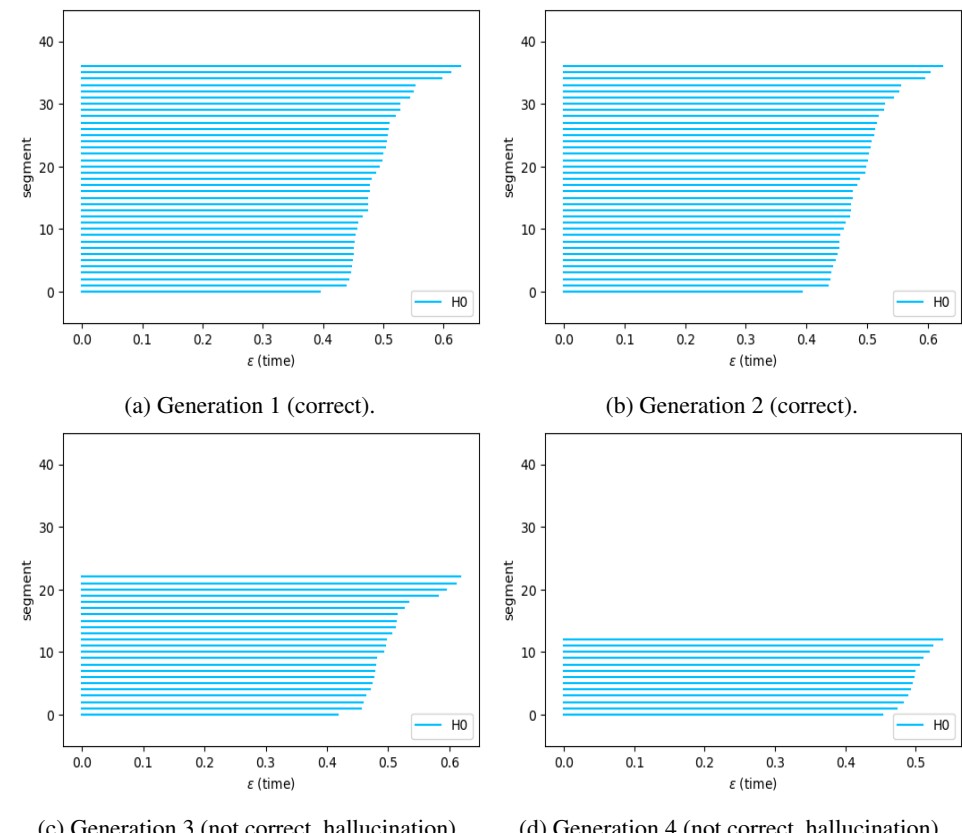

(a) Generation 1 (correct).

(b) Generation 2 (correct).

(c) Generation 3 (not correct, hallucination).

(d) Generation 4 (not correct, hallucination).

Figure 11: Examples of Cross-Barcode$_0$(P, G), CodeLLama-7B, HumanEval dataset, problem 14, layer 4, head 18. Cross-Barcode$_1$(P, G) is empty for these attention maps.

attention matrices of size (seq_len_k)$^2$ for a model with n_layers and n_heads for a dataset of size N using the formula: n_layers $\times$ n_heads $\times$ s $\times \sum_{i=1}^{k}$ (seq_len_k)$^2$ where s is the size of the float type. We assume s = 4 bytes. If one uses only attention matrices from the last layer of the model, we obtain the memory footprint approximately 20.7 - 31.1 Gb for the Human Eval dataset and 36.6 - 53.7 Gb for MBPP (depending on the model). However, to store the attention matrices for all layers and all heads, the memory footprint is about 578.2 - 996.4 Gb for Human Eval and 1023.5 - 1718.7 Gb for MBPP. We highlight that even for datasets of moderate size (i.e. 4100 generations for HumanEval and 2500 generations for MBPP), the memory footprint becomes prohibitively high. Thus, it is not always feasible to store such features. In contrast, in our approach, we do not need to store the attention matrices since we compute all the features immediately during generation. Hence, the size of our training dataset is negligible.

We provide an estimate of computational time using the CodeLlama-7B model and HumanEval dataset as an example. The average time taken to generate a solution for one problem without feature extraction is 6,2 sec, with feature extraction 11,5 sec for all layers and heads, which are processed in parallel. Feature computation time during inference can be further decreased if only the most important features are used for classification: according to Section 5.5, it is possible to use only 5% of all attention features without significant loss of classification quality The average memory footprint is $\approx$190Mb for HumanEval and $\approx$544Mb for MBPP. This size is a small fraction of GPU footprint during generation. Moreover, our approach demonstrates high performance without hyperparameter tuning. Therefore, we believe that the proposed approach has better scalability and is more practical.

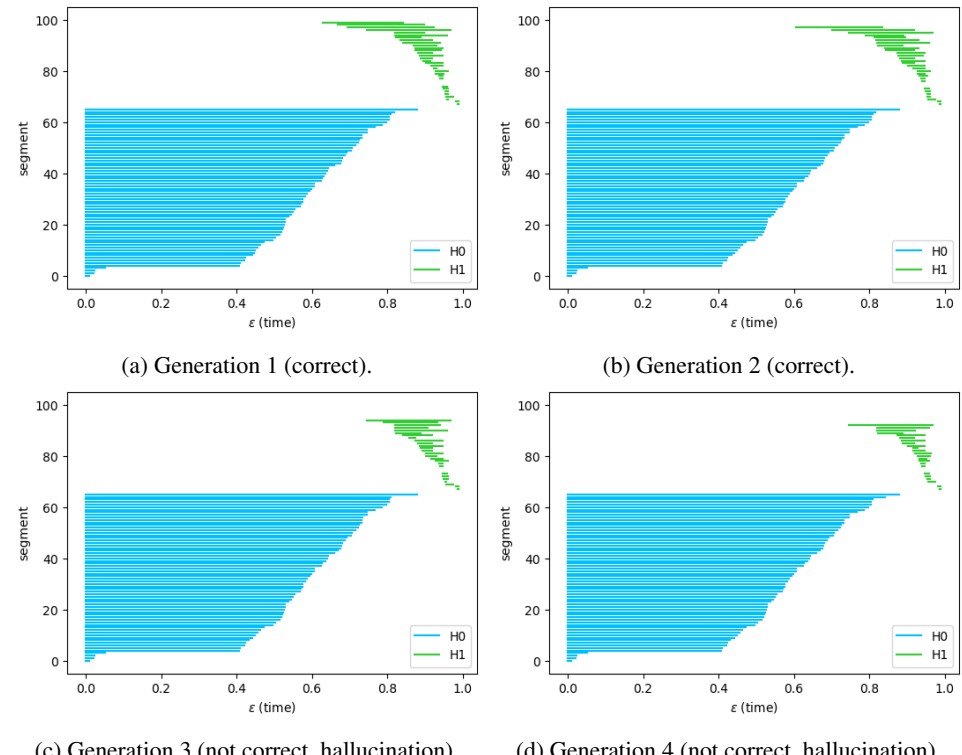

(a) Generation 1 (correct).

(b) Generation 2 (correct).

(c) Generation 3 (not correct, hallucination).

(d) Generation 4 (not correct, hallucination).

Figure 12: Examples of Cross-Barcode$_0$(G, P), Cross-Barcode$_1$(G, P). CodeLLama-7B, HumanEval dataset, problem 14, layer 15, head 5.

## G   ON DEFINITION OF A CODE HALLUCINATION

In our approach, the topological features obtained from attention maps account for the dissimilar structures in the prompt and generation subsets. Our intuition is that a correct solution should correspond to the structure of the prompt as their high-level semantic meanings correspond. Although other reasons behind hallucinations are possible, our approach estimates the correctness of code based only on the internal information flow of the model that does not require additional resources. Nevertheless, the proposed approach can be further integrated with other code hallucination detection tools to achieve better performance. Also, the most popular benchmarks like HumanEval and MBPP check only functional correctness, that is, whether a code solves the corresponding problem as stated in a prompt. Verification of a code is done by running functional tests, as explained in Section 5.1.

## H   LIMITATIONS

Although we have achieved good experimental results, we realize that our research has several limitations. Our method targets hallucinations that manifest in the geometry of the model's attention; it does not unravel all root causes (e.g., spurious pre-training correlations, decoding drift, RLHF bias). Extending the analysis to these factors is left for future work. We mostly explored code LLMs that have no more than 7B parameters. Information in larger models is more distributed in attention heads and results might differ. Also, processing more attention heads is computationally costly. Next, the proposed classifiers for hallucination detection are based on the attention maps of the same code LLMs as for code generations. We leave a more general setting for further research. Finally, our approach can predict whether a code is correct as a whole but can not point to a specific place with a bug.

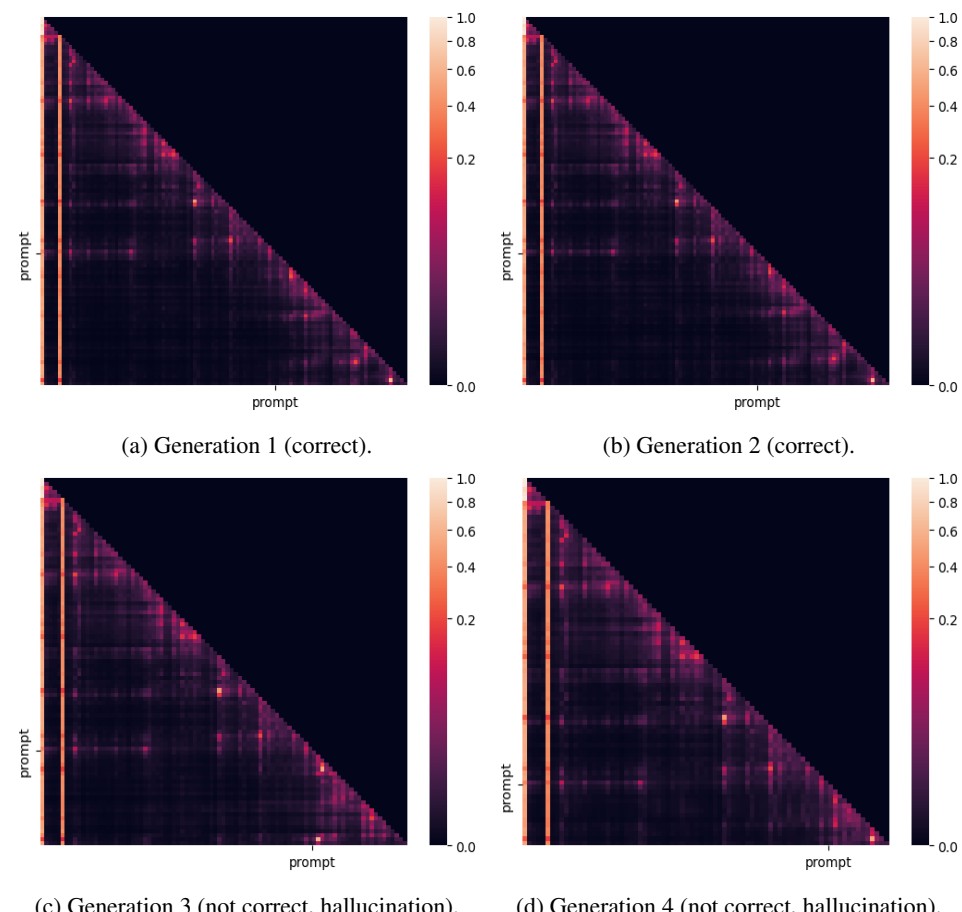

(a) Generation 1 (correct).

(b) Generation 2 (correct).

(c) Generation 3 (not correct, hallucination).

(d) Generation 4 (not correct, hallucination).

Figure 13: Attention maps. CodeLLama-7B, HumanEval dataset, problem 14, layer 4, head 18.

## I   ADDITIONAL ABLATION STUDY

Our main argument in the Ablation study is that different models benefit from different types of attention features. In this section, we provide an additional evaluation to further support our claim. We observe that the smaller QwenCoder-1.5b and QwenCoder-3b models experience a substantial performance decrease when topological features are removed; see Table 7. To further support our observations, we provide some examples of transferability from MBPP to HE datasets where the contribution of diagonal and MTD features is different for different models; see Table 8. Therefore, we propose to account for both types of features in our approach or at least to be aware of the potential benefits of using each type.

## J   EXPERIMENTS WITH LARGER MODELS

We carried out additional experiments, where code hallucinations are detected by a recent reasoning DeepSeek R1 model having 671 billion parameters with Chain of Thought inference on MBPP dataset. We used the following prompt:

*You are provided with two coding tasks with solutions. The first one is just an example and does not need an assessment. Tell whether the second task is correctly solved by the code provided in the second [BEGIN] [DONE] block. The answer must be Yes or No*

For code generated with the DeepSeek-6.7B model, we asked DeepSeek R1 using the above prompt, extracted the final answers (i.e. "Yes" or "No") from the generated responses, and trained the

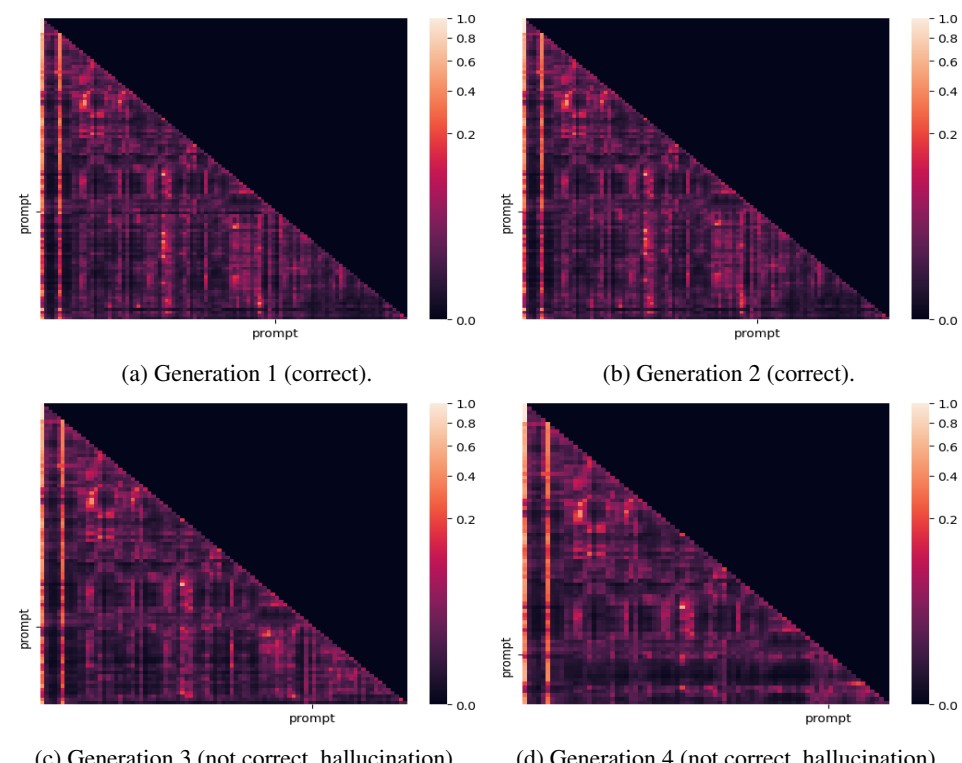

(a) Generation 1 (correct).

(b) Generation 2 (correct).

(c) Generation 3 (not correct, hallucination).

(d) Generation 4 (not correct, hallucination).

Figure 14: Attention maps. CodeLLama-7B, HumanEval dataset, problem 14, layer 15, head 5.

XGboost classifier using these features. The quality of hallucination detection via such zero-shot prompting is in Table 10, row "DeepSeek R1, (671B model)". The quality of the proposed approach is shown in Table 10, row "Attn. Feat (ours, from 6.7B model)". Note that in this case, we use *only* the attention features of the DeepSeek-6.7B model obtained during code generation. Finally, we combine both types of features to train a classifier and report its performance in Table 10, row "Attn. Feat. (ours, from 6.7B model) + DeepSeek R1". The larger DeepSeek R1 model demonstrates better performance than the classifier trained on attention features. However, by adding an output of DeepSeek R1 to our attention-based features and training the XGBoost classifier, we can achieve the best ROC-AUC score. The study of DeepSeek R1's Chain of Thoughts shows that this LLM is doing verification of code by interpreting Python code step by step for unit tests. This can explain the high accuracy of Deep Seek R1. At the same time, our method opens opportunities for a deeper understanding of inner working and information flow inside transformer models. Our attention features were based on a small 6.7B model in this experiment, however, our features were able to improve the performance of DeepSeek R1.

Additionally, we performed a similar experiment with QwenCoder2.5-32B. In this case, we use the same QwenCoder2.5-32B to generate code, extract attention features, and evaluate its performance with zero-shot prompting. Table 11 provides the experimental results. The proposed approach is capable of achieving a better detection quality than zero-shot the prompting, which supports the applicability of the proposed approach to larger models.

Furthermore, we provide a comparison with CODEJUDGE Tong & Zhang (2024), a code evaluation framework that uses LLM to analyze code functionality and then decide on code correctness. In our comparison, we use CodeLlama-Instruct 34B as an evaluation model to produce binary output to show whether the generated code is correct or not. We report the mean results over 3 runs in accordance with the original setup; see Tables 13, 14, 15.

We highlight that for a fair comparison we should consider the setup when CODEJUDGE is run without reference and Attn. Feat. is run for large 32-34B models. In our work, we did not include

Table 6: Ablation study for the choice of a classification model for code hallucination detection.

| Model | HumanEval | | MBPP | |
|---|---|---|---|---|
| | ROC-AUC | F1-Score | ROC-AUC | F1-Score |
| StarCoder2-7B | | | | |
| XGBoost | $82.9 \pm 2.7$ | $54.2 \pm 6.9$ | $81.9 \pm 2.4$ | $68.4 \pm 5.3$ |
| MLP | $80.9 \pm 4.7$ | $60.7 \pm 7.6$ | $81.1 \pm 1.7$ | $68.7 \pm 3.0$ |
| Logistic Regression | $84.3 \pm 3.9$ | $64.2 \pm 8.0$ | $\mathbf{82.9 \pm 2.3}$ | $\mathbf{69.7 \pm 3.4}$ |
| SVC | $\mathbf{84.9 \pm 4.0}$ | $\mathbf{65.6 \pm 7.4}$ | $82.3 \pm 2.6$ | $69.1 \pm 4.4$ |
| CodeLlama-7B | | | | |
| XGBoost | $\mathbf{85.6 \pm 3.9}$ | $56.4 \pm 7.2$ | $\mathbf{83.4 \pm 3.3}$ | $64.0 \pm 4.4$ |
| MLP | $81.8 \pm 7.2$ | $58.1 \pm 11.8$ | $81.6 \pm 2.2$ | $\mathbf{64.4 \pm 5.6}$ |
| Logistic Regression | $81.8 \pm 7.1$ | $58.2 \pm 7.9$ | $82.6 \pm 2.1$ | $63.6 \pm 4.7$ |
| SVC | $83.2 \pm 5.4$ | $\mathbf{59.8 \pm 9.2}$ | $82.4 \pm 1.2$ | $60.9 \pm 4.6$ |
| DeepSeek-Coder-6.7B | | | | |
| XGBoost | $85.6 \pm 2.8$ | $68.9 \pm 5.5$ | $\mathbf{82.6 \pm 1.9}$ | $\mathbf{76.5 \pm 2.7}$ |
| MLP | $84.3 \pm 2.4$ | $69.6 \pm 3.9$ | $81.7 \pm 1.1$ | $74.8 \pm 1.9$ |
| Logistic Regression | $85.8 \pm 3.2$ | $71.1 \pm 4.7$ | $81.8 \pm 2.4$ | $75.7 \pm 2.2$ |
| SVC | $\mathbf{87.0 \pm 2.4}$ | $\mathbf{72.9 \pm 3.7}$ | $81.4 \pm 1.6$ | $75.2 \pm 1.7$ |
| Qwen2.5-Coder-7B | | | | |
| XGBoost | $\mathbf{81.7 \pm 2.8}$ | $70.2 \pm 4.2$ | $\mathbf{82.2 \pm 2.2}$ | $75.4 \pm 1.7$ |
| MLP | $81.3 \pm 1.6$ | $70.8 \pm 2.5$ | $79.5 \pm 2.0$ | $72.9 \pm 3.0$ |
| Logistic Regression | $80.0 \pm 1.6$ | $\mathbf{71.3 \pm 3.8}$ | $81.0 \pm 1.9$ | $75.9 \pm 2.0$ |
| SVC | $79.6 \pm 1.7$ | $68.9 \pm 2.4$ | $81.0 \pm 1.7$ | $\mathbf{76.7 \pm 3.3}$ |
| Magicoder-S-DS-6.7B | | | | |
| XGBoost | $\mathbf{82.3 \pm 4.9}$ | $80.7 \pm 3.6$ | $77.8 \pm 2.5$ | $73.4 \pm 3.4$ |
| MLP | $76.5 \pm 3.3$ | $78.9 \pm 1.8$ | $78.6 \pm 3.6$ | $73.3 \pm 3.5$ |
| Logistic Regression | $81.7 \pm 1.6$ | $81.6 \pm 2.4$ | $81.3 \pm 2.8$ | $\mathbf{82.1 \pm 1.7}$ |
| SVC | $80.5 \pm 2.4$ | $\mathbf{82.0 \pm 3.1}$ | $\mathbf{82.5 \pm 2.6}$ | $81.3 \pm 1.9$ |

Table 7: HumanEval and MBPP features ablation for smaller models.

| Model | HumanEval | | MBPP | |
|---|---|---|---|---|
| | ROC-AUC | F1-Score | ROC-AUC | F1-Score |
| QwenCoder2.5-1.5b | | | | |
| Attn. Feat. (ours) | $\mathbf{85.9 \pm 4.1}$ | $\mathbf{58.1 \pm 9.6}$ | $\mathbf{82.8 \pm 1.4}$ | $\mathbf{59.2 \pm 2.7}$ |
| - w/o Diag. Feat. | $85.0 \pm 3.8$ | $57.6 \pm 6.8$ | $82.1 \pm 1.8$ | $58.6 \pm 4.2$ |
| - w/o MTD Feat. | $82.9 \pm 3.9$ | $53.6 \pm 10.8$ | $79.5 \pm 1.5$ | $57.0 \pm 4.5$ |
| QwenCoder2.5-3b | | | | |
| Attn. Feat. (ours) | $\mathbf{80.4 \pm 6.6}$ | $65.5 \pm 8.5$ | $78.0 \pm 3.0$ | $73.3 \pm 3.6$ |
| - w/o Diag. Feat. | $79.0 \pm 6.1$ | $\mathbf{67.2 \pm 6.4}$ | $\mathbf{79.6 \pm 3.0}$ | $\mathbf{74.8 \pm 3.9}$ |
| - w/o MTD Feat. | $74.9 \pm 4.4$ | $59.0 \pm 5.7$ | $75.5 \pm 2.2$ | $71.0 \pm 3.1$ |

reference (i.e. correct solution) to the prompt as this setup is not practical (typically, one does not know the correct solution). Also, since CODEJUDGE is evaluated with a 34B model, we evaluate the Attn. Feat. in a similar way to keep the experimental design of both methods as close as possible. In this comparison, Attn. Feat. outperforms CODEJUDGE in the majority of cases, winning by a large margin for ROC-AUC. However, to further explore the capabilities of the proposed Attn. Feat., we provide additional comparison when CODEJUDGE is run with reference and when Attn. Feat. is run for smaller 6.7B-7B models. In these cases, Attn. Feat. still outperforms CODEJUDGE as measured by ROC-AUC although the F1-score performance is lower in some cases.

Table 8: MBPP transferability ablation.

| Method | DeepSeek-Coder-6.7b | QwenCoder2.5-3b | StarCoder2-7b |
|---|---|---|---|
| Attn. Feat. (ours) | **72.4** | 65.6 | **67.2** |
| - w/o Diag. Feat. | 71.3 | **67.4** | 64.2 |
| - w/o MTD Feat. | 63.5 | 61.3 | 66.8 |

Table 9: HumanEval and MBPP features ablation

| Model | HumanEval | | MBPP | |
|---|---|---|---|---|
| | ROC-AUC | F1-Score | ROC-AUC | F1-Score |
| StarCoder2-7B | | | | |
| Attn. Feat. (ours) | $82.9 \pm 2.7$ | $54.2 \pm 6.9$ | $\mathbf{81.9 \pm 2.4}$ | $\mathbf{68.4 \pm 5.3}$ |
| - w/o Diag. Feat. | $82.2 \pm 4.5$ | $\mathbf{56.1 \pm 9.7}$ | $80.5 \pm 2.8$ | $66.3 \pm 5.3$ |
| - w/o MTD Feat. | $\mathbf{83.8 \pm 2.7}$ | $52.5 \pm 8.4$ | $81.1 \pm 2.6$ | $67.7 \pm 5.0$ |
| CodeLlama-7B | | | | |
| Attn. Feat. (ours) | $\mathbf{85.6 \pm 3.9}$ | $56.4 \pm 7.2$ | $83.4 \pm 2.2$ | $\mathbf{64.0 \pm 4.4}$ |
| - w/o Diag. Feat. | $83.5 \pm 4.8$ | $50.0 \pm 6.5$ | $81.5 \pm 2.6$ | $60.2 \pm 4.2$ |
| - w/o MTD Feat. | $85.5 \pm 4.4$ | $\mathbf{58.3 \pm 10.1}$ | $\mathbf{83.5 \pm 1.8}$ | $63.9 \pm 4.3$ |
| DeepSeek-Coder-6.7B | | | | |
| Attn. Feat. (ours) | $\mathbf{85.6 \pm 2.8}$ | $\mathbf{68.9 \pm 5.5}$ | $\mathbf{82.6 \pm 1.9}$ | $\mathbf{76.5 \pm 2.7}$ |
| - w/o Diag. Feat. | $85.1 \pm 2.2$ | $67.0 \pm 5.9$ | $81.3 \pm 2.6$ | $74.9 \pm 3.2$ |
| - w/o MTD Feat. | $84.4 \pm 2.2$ | $67.1 \pm 3.8$ | $82.2 \pm 1.7$ | $75.9 \pm 1.7$ |
| Qwen2.5-Coder-7B | | | | |
| Attn. Feat. (ours) | $\mathbf{81.7 \pm 2.8}$ | $\mathbf{70.2 \pm 4.2}$ | $\mathbf{82.2 \pm 2.2}$ | $\mathbf{75.4 \pm 1.7}$ |
| - w/o Diag. Feat. | $80.6 \pm 2.3$ | $68.9 \pm 3.9$ | $80.8 \pm 2.1$ | $\mathbf{75.4 \pm 0.4}$ |
| - w/o MTD Feat. | $78.9 \pm 1.9$ | $66.4 \pm 1.4$ | $76.9 \pm 2.2$ | $71.6 \pm 1.7$ |
| Magicoder-S-DS-6.7B | | | | |
| Attn. Feat. (ours) | $\mathbf{82.3 \pm 4.9}$ | $80.7 \pm 3.6$ | $\mathbf{81.5 \pm 1.6}$ | $\mathbf{80.6 \pm 2.0}$ |
| - w/o Diag. Feat. | $79.8 \pm 2.7$ | $81.1 \pm 1.8$ | $81.2 \pm 1.9$ | $80.1 \pm 1.4$ |
| - w/o MTD Feat. | $82.1 \pm 3.4$ | $\mathbf{81.6 \pm 2.6}$ | $80.3 \pm 2.5$ | $79.7 \pm 2.2$ |

## K  ADDITIONAL RESULTS ON CONTRIBUTION OF INDIVIDUAL HEADS

We carry out additional experiments with 7B-models and MultiPL-E benchmark [9], which is a translation of HumanEval to several popular programming languages; we used Go, Java, Rust, Lua among them. We find that features of some heads have quite a high correlation with the target value (presence of a hallucination) and can be used as individual predictors. In Table 17 we report the ROC-AUC scores of the top-performing features.

## L  EVALUATION WITH GREEDY DECODING

In the main experiments in Section 5.2, we use sampling with temperature 0.8 to generate diverse solutions for each coding problem and obtain a larger train and test samples. However, production systems typically use greedy decoding, and in this section we fill this gap. We evaluate the performance of the proposed Attn. Feat. classifier when both train and test data are generated via greedy decoding. In this setup, the sample sizes are 164 for HumanEval and 500 for MBPP with 1 generation per task, and we use cross-validation with 5 folds. Although the sample size is sufficiently smaller than in Section 5.2, the proposed classifier achieves high performance across all models and datasets and can be applied to small size samples; see Table 21.

---

[9]https://github.com/nuprl/MultiPL-E

Table 10: MBPP features for larger models. See Section J for details.

| Method | ROC-AUC |
|---|---|
| DeepSeek-Coder-6.7B | |
| Attn. Feat (ours, from 6.7B model) | $82.6 \pm 1.9$ |
| DeepSeek R1 (671B model) | $92.3 \pm 1.1$ |
| Attn. Feat. (from 6.7B model) + DeepSeek R1 (671B) | $95.1 \pm 0.7$ |

Table 11: HumanEval features for larger models. See Section J for details.

| Method | ROC-AUC | F1-Score |
|---|---|---|
| QwenCoder2.5-32B | | |
| Attn. Feat (ours) | $85.0 \pm 2.7$ | $77.0 \pm 5.5$ |
| Zero-shot prompt | $67.4 \pm 3.5$ | $71.8 \pm 3.6$ |

## M  CROSS-MODEL TRANSFERABILITY

In this section, we verify whether the proposed XGBoost classifier trained on the attention features of one model can be effectively applied to detect hallucinations using the attenton features of another model. A formal requirement is that the number of attention features of the two models should be the same. The results of cross-model transferability are shown in Tables 22, 23. We conclude that a classifier trained on DeepSeek-Coder-6.7B is transferable to Magicoder-S-DS-6.7B and vice versa, which is not the case for CodeLlama-7B. We suppose the reason is that Magicoder-S-DS-6.7B is a fine-tuned DeepSeek-Coder-6.7B and a correspondence between attention heads persists.

## N  FAILURE CASE STUDY

We provide a failure case study in the Table 24. For each trained classifier, we analyze the category of generated code which was classified incorrectly by the classifier. We report the fraction of top-3 most popular code categories w.r.t. number of samples in the test sample. We use 5-fold stratified group cross validation and report the average values over 5 folds. The table reveals the most popular failure cases: misclassification of correct codes that passed the tests (referred to as 'passed') and misclassification of wrong solutions (i.e. codes that did not pass any test and caused an AssertionError). The fraction of other errors (top-3 and others) is sufficiently lower.

Table 12: Code hallucination detection for HumanEval dataset for larger models. For each task, 10 candidate solutions were generated. **Bold** and underline denote the first and second best results.

| Method | ROC-AUC | F1-Score |
|---|---|---|
| CodeLlama-34B | | |
| Prompt Len. | $57.3 \pm 6.3$ | $37.7 \pm 13.6$ |
| Gen. Len. | $62.8 \pm 1.8$ | $38.2 \pm 4.0$ |
| Mean Log. Prob. | $74.1 \pm 5.0$ | $51.8 \pm 7.6$ |
| CodeT5-base ft. | $54.7 \pm 5.5$ | $0.0 \pm 0.0$ |
| Attn. Feat. (ours) | $83.2 \pm 3.8$ | $62.5 \pm 7.1$ |
| All. Feat. | **$84.8 \pm 2.8$** | **$62.8 \pm 6.7$** |
| Qwen2.5-Coder-32B | | |
| Prompt Len. | $53.2 \pm 9.2$ | $53.4 \pm 12.8$ |
| Gen. Len. | $58.0 \pm 3.1$ | $62.5 \pm 3.8$ |
| Mean Log. Prob. | $65.8 \pm 4.8$ | $63.6 \pm 2.8$ |
| CodeT5-base ft. | $53.3 \pm 4.5$ | $59.5 \pm 15.2$ |
| Attn. Feat. (ours) | **$85.0 \pm 2.7$** | $77.0 \pm 5.5$ |
| All. Feat. | $84.9 \pm 3.1$ | **$77.7 \pm 5.7$** |
| DeepSeek-Coder-33B | | |
| Prompt Len. | $53.3 \pm 6.5$ | $44.0 \pm 5.3$ |
| Gen. Len. | $56.6 \pm 4.2$ | $48.2 \pm 3.4$ |
| Mean Log. Prob. | $66.4 \pm 3.4$ | $57.1 \pm 2.9$ |
| CodeT5-base ft. | $57.6 \pm 3.9$ | $1.1 \pm 2.2$ |
| Attn. Feat. (ours) | $88.0 \pm 2.9$ | $77.9 \pm 2.2$ |
| All. Feat. | **$88.9 \pm 2.8$** | **$78.6 \pm 3.1$** |

Table 13: Additional comparison with CODEJUDGE Tong & Zhang (2024) on HumanEval for 32-34B models.

| Method | CodeLlama-34B | | DeepSeek-Coder-33B | | Qwen2.5-Coder-32B | |
|---|---|---|---|---|---|---|
| | ROC-AUC | F1-Score | ROC-AUC | F1-Score | ROC-AUC | F1-Score |
| CODEJUDGE A.S. w/o reference | $65.3 \pm 3.0$ | $69.6 \pm 4.8$ | $72.0 \pm 7.2$ | $80.2 \pm 3.8$ | $61.6 \pm 7.3$ | $81.8 \pm 4.7$ |
| CODEJUDGE A.S. w/ reference | $68.7 \pm 3.8$ | $70.0 \pm 5.4$ | $75.8 \pm 5.6$ | $81.1 \pm 2.4$ | $62.0 \pm 7.5$ | $80.3 \pm 2.0$ |
| Attn. Feat. | $75.0 \pm 5.1$ | $68.4 \pm 5.9$ | $86.5 \pm 7.2$ | $81.6 \pm 4.0$ | $74.0 \pm 6.4$ | $82.7 \pm 1.6$ |

Table 14: Additional comparison with CODEJUDGE Tong & Zhang (2024) on HumanEval for Star-Coder2-7B, CodeLlama-7B, DeepSeekCoder-6.7B.

| Method | StarCoder2-7B | | CodeLlama-7B | | DeepSeekCoder-6.7b | |
|---|---|---|---|---|---|---|
| | ROC-AUC | F1-Score | ROC-AUC | F1-Score | ROC-AUC | F1-Score |
| CODEJUDGE A.S. w/o reference | $70.6 \pm 4.7$ | $64.8 \pm 4.3$ | $71.3 \pm 4.1$ | $59.2 \pm 3.8$ | $69.2 \pm 3.2$ | $74.4 \pm 4.8$ |
| CODEJUDGE A.S. w/ reference | $75.8 \pm 4.7$ | $68.2 \pm 3.9$ | $73.0 \pm 3.3$ | $61.0 \pm 5.3$ | $68.8 \pm 7.9$ | $70.9 \pm 9.0$ |
| Attn. Feat. | $80.8 \pm 3.7$ | $63.5 \pm 9.1$ | $80.1 \pm 3.9$ | $59.9 \pm 9.4$ | $83.9 \pm 8.4$ | $78.5 \pm 6.7$ |

Table 15: Additional comparison with CODEJUDGE Tong & Zhang (2024) on HumanEval for QwenCoder-7B, Magicoder-S-DS-6.7B.

| Method | QwenCoder-7b | | Magicoder-S-DS-6.7b | |
|---|---|---|---|---|
| | ROC-AUC | F1-Score | ROC-AUC | F1-Score |
| CODEJUDGE A.S. w/o reference | $67.2 \pm 5.7$ | $78.1 \pm 3.5$ | $55.3 \pm 2.7$ | $83.3 \pm 6.6$ |
| CODEJUDGE A.S. w/ reference | $70.4 \pm 7.4$ | $77.6 \pm 4.3$ | $57.9 \pm 7.1$ | $82.1 \pm 3.6$ |
| Attn. Feat. | $73.3 \pm 7.1$ | $73.7 \pm 6.1$ | $65.1 \pm 3.4$ | $85.4 \pm 5.4$ |

Table 16: Code hallucination detection with attention features for MBPP dataset with increasing complexity of prompt: two-shot prompts.

| Method | ROC-AUC | F1-Score |
|---|---|---|
| Qwen2.5-Coder-7B | | |
| Prompt Len. | $53.5 \pm 2.9$ | $60.2 \pm 1.5$ |
| Gen. Len. | $56.9 \pm 3.3$ | $63.2 \pm 2.0$ |
| Mean Log. Prob. | $58.1 \pm 2.6$ | $61.1 \pm 3.3$ |
| Attn. Feat. (ours) | $78.3 \pm 3.1$ | $74.2 \pm 1.8$ |
| All. Feat. | $76.7 \pm 2.7$ | $72.2 \pm 2.1$ |
| Self-Eval | $68.4 \pm 1.0$ | $77.4 \pm 2.4$ |
| Interrogate-LLM | $56.6 \pm 1.5$ | $69.1 \pm 2.1$ |
| DeepSeek-Coder-6.7B | | |
| Prompt Len. | $56.0 \pm 2.0$ | $58.0 \pm 2.4$ |
| Gen. Len. | $54.4 \pm 1.3$ | $59.6 \pm 1.6$ |
| Mean Log. Prob. | $64.9 \pm 1.7$ | $64.2 \pm 2.4$ |
| Attn. Feat. (ours) | $81.4 \pm 2.2$ | $75.0 \pm 5.3$ |
| All. Feat. | $80.4 \pm 0.7$ | $74.5 \pm 2.6$ |
| Self-Eval | $50.0 \pm 0.0$ | $69.3 \pm 5.1$ |
| Interrogate-LLM | $61.1 \pm 3.1$ | $69.4 \pm 5.2$ |

Table 17: ROC AUC scores of top-performing features across several benchmarks.

| Feature | HumalEval | | | | | MBPP |
|---|---|---|---|---|---|---|
| | Python | Go | Rust | Java | Lua | Python |
| StarCoder2-7B | | | | | | |
| avg. prompt's self-attention, layer 14, head 0 | 70.8 | 76.8 | 74.6 | 68.9 | 70.3 | 55.2 |
| avg. prompt's self-attention, layer 15, head 5 | 71.1 | 78.4 | 75.1 | 72.9 | 72.4 | 58.1 |
| avg. prompt's self-attention, layer 23, head 20 | 69.2 | 72.2 | 74.2 | 64.7 | 67.7 | 50.7 |
| CodeLlama-7B | | | | | | |
| $-MTD_1(P, G)/\|P\|$, layer 15, head 27 | 67.6 | 71.1 | 73.7 | 71.0 | – | 66.1 |
| avg. prompt's self-attention, layer 7, head 22 | 74.6 | 75.7 | 76.5 | 69.7 | – | 63.1 |
| $MTD_0(P, G)/\|P\|$, layer 11, head 23 | 69.1 | 71.0 | 71.9 | 65.4 | – | 69.1 |
| DeepSeek-Coder-6.7B | | | | | | |
| $MTD_0(P, G)/\|P\|$, layer 30, head 11 | 67.5 | 65.7 | 67.0 | 66.3 | 69.1 | 58.1 |
| avg. prompt's self-attention, layer 12, head 17 | 65.0 | 73.6 | 70.0 | 69.8 | 69.8 | 58.2 |
| avg. prompt's self-attention, layer 12, head 23 | 64.1 | 71.0 | 66.8 | 64.4 | 67.6 | 58.2 |
| Qwen2.5-Coder-7B | | | | | | |
| avg. generation's self-attention, layer 24, head 2 | 65.8 | 60.2 | 58.8 | 60.2 | 66.7 | 60.8 |
| $MTD_0(P, G)/\|P\|$, layer 11, head 5 | 66.2 | 68.8 | 59.5 | 61.6 | 72.7 | 59.4 |
| avg. prompt's self-attention, layer 9, head 26 | 65.4 | 75.8 | 66.9 | 67.0 | 71.1 | 61.0 |
| Magicoder-S-DS-6.7B | | | | | | |
| $MTD_0(P, G)/\|P\|$, layer 30, head 11 | 68.8 | 70.2 | 60.0 | 65.1 | 63.8 | 58.2 |
| avg. prompt's self-attention, layer 13, head 13 | 63.1 | 64.9 | 69.0 | 67.3 | 63.2 | 58.8 |
| avg. prompt's self-attention, layer 12, head 17 | 63.0 | 63.2 | 68.0 | 67.1 | 60.8 | 58.9 |

Table 18: Characteristics of generated data, Pass@1. For each problem, we generated 25 candidate solutions for HumanEval, Python, 10 candidate solutions for HumanEval, Java, Go, Rust, Lua, and 5 candidate solutions for MBPP.

| Model | HumalEval | | | | | MBPP |
|---|---|---|---|---|---|---|
| | Python | Java | Go | Rust | Lua | Python |
| StarCoder2-7B | 28.9 | 24.5 | 17.5 | 20.9 | 19.1 | 42.8 |
| CodeLlama-7B | 25.9 | 25.8 | 17.6 | 20.8 | 0.0 | 35.2 |
| DeepSeek-Coder-6.7B | 40.3 | 33.5 | 23.6 | 28.7 | 16.6 | 52.6 |
| Qwen2.5-Coder-7B | 47.8 | 22.7 | 11.9 | 22.6 | 23.5 | 52.1 |
| Magicoder-S-DS-6.7B | 65.5 | 48.9 | 40.3 | 44.2 | 34.6 | 61.3 |

Table 19: Performance of the proposed method on different programming languages, ROC-AUC.

| Model | Java | Go | Rust | Lua |
|---|---|---|---|---|
| StarCoder2-7B | $82.7 \pm 4.9$ | $86.5 \pm 3.6$ | $82.4 \pm 5.2$ | $82.0 \pm 3.5$ |
| CodeLlama-7B | $76.8 \pm 6.1$ | $81.9 \pm 6.2$ | $77.5 \pm 10.8$ | – |
| DeepSeek-Coder-6.7B | $84.9 \pm 4.2$ | $84.7 \pm 2.9$ | $82.8 \pm 7.1$ | $86.7 \pm 4.3$ |
| Qwen2.5-Coder-7B | $82.6 \pm 4.1$ | $91.8 \pm 0.9$ | $87.3 \pm 2.6$ | $90.2 \pm 3.0$ |
| Magicoder-S-DS-6.7B | $77.8 \pm 4.5$ | $80.8 \pm 2.0$ | $75.1 \pm 5.9$ | $79.0 \pm 3.2$ |

Table 20: Performance of the proposed method on different programming languages, F1-Score.

| Model | Java | Go | Rust | Lua |
|---|---|---|---|---|
| StarCoder2-7B | $50.4 \pm 7.6$ | $39.7 \pm 11.9$ | $36.5 \pm 9.2$ | $39.0 \pm 8.7$ |
| CodeLlama-7B | $36.4 \pm 20.6$ | $26.5 \pm 10.2$ | $34.8 \pm 17.3$ | – |
| DeepSeek-Coder-6.7B | $64.1 \pm 4.5$ | $53.2 \pm 8.7$ | $56.8 \pm 6.6$ | $41.6 \pm 15.1$ |
| Qwen2.5-Coder-7B | $45.8 \pm 9.6$ | $37.6 \pm 11.5$ | $49.9 \pm 6.9$ | $64.6 \pm 7.9$ |
| Magicoder-S-DS-6.7B | $68.0 \pm 4.9$ | $63.3 \pm 2.0$ | $63.1 \pm 8.7$ | $58.4 \pm 6.7$ |

Table 21: Quality of code hallucination detection for HumanEval and MBPP datasets when greedy decoding is used to generate cadidate solutions.

| Model | HumalEval | | MBPP | |
|---|---|---|---|---|
| | ROC-AUC | F1-Score | ROC-AUC | F1-Score |
| StarCoder2-7B | $80.8 \pm 3.7$ | $61.9 \pm 9.5$ | $79.3 \pm 6.1$ | $69.7 \pm 5.4$ |
| CodeLlama-7B | $80.1 \pm 3.9$ | $59.4 \pm 10.0$ | $82.3 \pm 3.1$ | $71.3 \pm 6.3$ |
| DeepSeek-Coder-6.7B | $83.9 \pm 8.4$ | $78.5 \pm 6.7$ | $79.6 \pm 3.4$ | $78.7 \pm 2.1$ |
| Qwen2.5-Coder-7B | $73.3 \pm 7.1$ | $70.1 \pm 6.4$ | $74.0 \pm 1.9$ | $75.7 \pm 4.0$ |
| Magicoder-S-DS-6.7B | $65.1 \pm 3.4$ | $82.6 \pm 4.3$ | $79.3 \pm 2.0$ | $80.8 \pm 1.3$ |

Table 22: Quality of code hallucination detection in cross-model transferability on HumanEval. Rows correspond to train and columns correspond test samples.

| Test | CodeLlama-7B | | DeepSeek-Coder-6.7B | | Magicoder-S-DS-6.7B | |
|---|---|---|---|---|---|---|
| Train | ROC-AUC | F1-Score | ROC-AUC | F1-Score | ROC-AUC | F1-Score |
| CodeLlama-7B | $85.6 \pm 3.9$ | $56.4 \pm 7.2$ | $51.9 \pm 13.2$ | $17.3 \pm 21.1$ | $51.1 \pm 7.1$ | $24.3 \pm 29.0$ |
| DeepSeek-Coder-6.7B | $45.4 \pm 10.3$ | $14.2 \pm 9.6$ | $85.6 \pm 2.8$ | $68.9 \pm 5.5$ | $79.4 \pm 3.7$ | $69.4 \pm 6.8$ |
| Magicoder-S-DS-6.7B | $51.8 \pm 5.4$ | $23.4 \pm 17.7$ | $83.0 \pm 2.2$ | $70.7 \pm 2.7$ | $82.3 \pm 4.9$ | $80.7 \pm 3.6$ |

Table 23: Quality of code hallucination detection in cross-model transferability on MBPP. Rows correspond to train and columns correspond test samples.

| Test | CodeLlama-7B | | DeepSeek-Coder-6.7B | | Magicoder-S-DS-6.7B | |
|---|---|---|---|---|---|---|
| Train | ROC-AUC | F1-Score | ROC-AUC | F1-Score | ROC-AUC | F1-Score |
| CodeLlama-7B | $83.4 \pm 3.3$ | $64.0 \pm 4.4$ | $49.3 \pm 2.9$ | $17.2 \pm 22.6$ | $52.3 \pm 9.4$ | $21.9 \pm 26.8$ |
| DeepSeek-Coder-6.7B | $49.2 \pm 5.5$ | $21.0 \pm 17.8$ | $82.6 \pm 1.9$ | $76.5 \pm 2.7$ | $75.2 \pm 1.6$ | $54.5 \pm 11.3$ |
| Magicoder-S-DS-6.7B | $51.8 \pm 3.4$ | $29.2 \pm 24.0$ | $64.8 \pm 6.4$ | $58.5 \pm 13.9$ | $81.5 \pm 1.6$ | $80.6 \pm 2.0$ |

Table 24: Failure case study. "P" - Passed, "AE" - AssertionError, "NE" - NameError, "IE" - Index-Error, "TE" - TypeError.

| Model | HumanEval | | | MBPP | | |
|---|---|---|---|---|---|---|
| | Top-1 | Top-2 | Top-3 | Top-1 | Top-2 | Top-3 |
| StarCoder2-7b | P $17.2 \pm 4.5$ | AE $2.9 \pm 2.2$ | NE $0.8 \pm 0.5$ | P $15.0 \pm 3.5$ | AE $9.7 \pm 2.2$ | NE $0.7 \pm 0.4$ |
| CodeLLama-7b | P $14.6 \pm 3.6$ | AE $3.8 \pm 2.7$ | NE $0.2 \pm 0.2$ | P $15.5 \pm 1.6$ | AE $7.4 \pm 1.2$ | NE $0.5 \pm 0.4$ |
| DeepSeekCoder-6.7b | P $14.6 \pm 2.9$ | AE $4.9 \pm 1.2$ | IE $0.5 \pm 0.6$ | AE $12.6 \pm 3.2$ | P $11.1 \pm 3.8$ | NE $0.7 \pm 0.5$ |
| QwenCoder-7b | P $16.0 \pm 5.4$ | AE $6.9 \pm 2.8$ | NE $2.1 \pm 0.7$ | P $11.3 \pm 0.5$ | AE $10.8 \pm 1.4$ | NE $1.3 \pm 0.5$ |
| Magicoder-6.7b | AE $12.1 \pm 4.8$ | P $10.4 \pm 4.8$ | IE $0.7 \pm 0.6$ | AE $14.5 \pm 4.4$ | P $9.2 \pm 2.3$ | TE $0.6 \pm 0.5$ |

