# OpenReview forum: "Topology of Attention Detects Hallucinations in Code LLMs"
_ICLR.cc/2026/Conference — ICLR 2026 Conference Withdrawn Submission_

### Official Review · Reviewer_6RBV · 2025-10-27

**Soundness:** 2
**Presentation:** 2
**Contribution:** 2
**Rating:** 2
**Confidence:** 5

**Summary:**

This paper introduces a novel approach to detect hallucinations in code generated by Large Language Models using topological data analysis of attention maps. The authors transform attention patterns into weighted graphs by applying Manifold Topology Divergence (MTD) on the weighted graphs derived from the attention maps. These attention features are used to detect the hallucinations using a XGBoost algorithm. The paper claims that the proposed method is better than several baselines.

**Strengths:**

- The paper shows an innovative application of topological data analysis to code hallucination detection
- The paper evaluated the experiments on multiple code based LLMs, code benchmarks, and multiple languages

**Weaknesses:**

- "To enable diversity of generated solutions, a sampling with non-zero temperature
of was done." This sentence is unclear.
- The presentation in the paper can be better, there are a few grammatical errors, and the paper references a number of results from Appendix, even before explaining the results in the main body. Whenever the main body results are referenced, there is a lame reference saying the results are in so and so table, without actually explaining what the insights are from the observations from the experiments.
- The paper poses to use XGBoost as a classifier (I feel more as a meta classifier) on the attention features extracted from the transformer. There are number of clarifications sorted here:
  - Which attention layer are these coming from there is more than one attention layer in any given coding LLM used in the paper?
  - Why XGBoost as a classifier for hallucination detection?
  - Why XGBoost was compared against a Random classifier baseline? No details are offered about that Random baseline.
  - Why a random baseline, since the extracted features are already in the graph format, whynot use a simple light-weight Graphical Neural Net (GNN) based classifier as a baseline or even as the main classifier?
- What is the definition of the mean log probabilities of the generated tokens? Is the log applied on the softmax probabilities of the generated tokens?
- The paper does not talk about the computational overhead incurred in terms of the topological feature extraction.

**Questions:**

See the weaknesses section above for questions.

---

> ### Author Response · Authors · 2025-11-21
> **Response to Reviewer 6RBV, part 1/2**
>
> Thank you for your feedback. Below we respond to the questions one by one.
>
> **W1**. _"To enable diversity of generated solutions, a sampling with non-zero temperature of was done." This sentence is unclear._
>
> **A**: Thank you for your remark. We provide the correct version: "To enable diversity of generated solutions, a temperature sampling was done." Due to the limited size of the common benchmarks (164 problems in HumanEval and 500 problems in MBPP), we use temperature sampling to obtain diverse generations and increase the sample size. In addition, this setup allows us to perform the ranking experiment in Section 5.3. We provide details on the generation procedure in Appendix A.
>
> **W2**. _The presentation in the paper can be better, there are a few grammatical errors, and the paper references a number of results from Appendix, even before explaining the results in the main body. Whenever the main body results are referenced, there is a lame reference saying the results are in so and so table, without actually explaining what the insights are from the observations from the experiments._
>
> **A**: Thank you for your remark. We improve the writing style and provide a more detailed analysis of the experiments in the revision of the paper. Below we cover the changes made.
>
> Line 265-266: We add supplementary discussion: ``Tables 14, 16, 17 demonstrate that the performance of the proposed approach is comparable to performance in the main experiments, and the proposed Attn. Feat. can detect hallucinations even in low resource and niche languages. Moreover, there are several separate features that perform robust across the languages which further supports the importance of the proposed features.’’
>
> Line 266-267: We add supplementary discussion: ``Also, we increase the linguistic complexity of the prompt via two-shot prompts on MBPP dataset against one-shot prompt MBPP study in the main part. Table 13 demonstrates that the proposed Attn. Feat. classifier has consistently high quality and outperforms other baselines.‘’
>
> Line 323: We add supplementary discussion: ``This result justifies that the proposed Att. Feat. classifiers can distinguish different levels of code correctness among several candidate solutions to the same problem. Even for more recent Qwen2.5-Coder-7B and Magicoder-S-DS-6.7B, pass@1 can still be further improved with the use of Attn. Feat. classifier.’’
>
> Line 449-450: We add supplementary discussion: ``This result demonstrates that the proposed attention features are capable of identifying the majority of errors of different types, consistently across the models.’’
>
> **W3.1**. _Which attention layer are these coming from there is more than one attention layer in any given coding LLM used in the paper?_
>
> **A**: In the base setup (Tables 2,3,4) we compute features over all layers and heads of LLM and then train XGBoost on these features. In Section 5.5 we demonstrate that it is possible to use only 5\% of all features without a significant decrease in performance. Moreover, in Section K in the Appendix we analyze the quality of hallucination detection when only one feature is used. In Table 14, we provide several top-performing features. Although the quality is lower than from a XGBoost classifier with a full set of features, it still outperforms several baselines. We highlight that these results are obtained with only one feature used, but provide stable performance across multiple languages and problems.
>
> **W3.2**. _Why XGBoost as a classifier for hallucination detection?_
>
> **A**: We substantiate the choice of XGBoost in Section E in the Appendix. We provide an ablation study where we compare the quality of XGBoost classification with MLP, Logistic Regression, and SVC (Table 6) as other possible options for the lightweight classifier over the proposed attention features. Although some other classifiers may perform better in certain cases, we choose XGBoost as it offers (a) strong average performance across all the code LLMs, (b) negligible training cost (≈ 30s per fold), and (c) no hyper-parameter tuning in our setting. XGBoost guarantees low computational overhead while providing a single, robust baseline for subsequent work.
>
> **W3.3**. _Why XGBoost was compared against a Random classifier baseline? No details are offered about that Random baseline._
>
> **A**: In Section 5.3, we analyze how the trained classifier can be used to rank the candidates generated via temperature sampling. When computing pass@1, the standard approach is to pick a code candidate randomly and execute it against the provided tests. We demonstrate that choosing the code candidates according to classifier probabilities improves pass@1 and can be further used to prioritize generated solutions.

---

> > ### Author Response · Authors · 2025-11-21
> > **Response to Reviewer 6RBV, part 2/2**
> >
> > **W3.4** _Why a random baseline, since the extracted features are already in the graph format, why not use a simple light-weight Graphical Neural Net (GNN) based classifier as a baseline or even as the main classifier?_
> >
> > **A**: We discuss applicability of GNNs to attention maps in Section F of the Appendix. We represent the attention map as a graph and then compute the proposed features. However, to train a lightweight GNN on attention maps represented as graphs, we need to store them. Memory consumption may be too high and not possible in practice (see Appendix F). Alternatively, we empirically show that attention maps contain valuable topological information for hallucination detection and propose a more effective way to incorporate them into the classification pipeline with a lower memory cost.
> >
> > **W4**. _What is the definition of the mean log probabilities of the generated tokens? Is the log applied on the softmax probabilities of the generated tokens?_
> >
> > **A**: We obtain the logits of tokens and apply softmax to get probability distributions over vocabulary. Then we split all the tokens into tokens of prompt and tokens of generation and apply the mean logarithm of corresponding probabilities.
> >
> > **W5**. _The paper does not talk about the computational overhead incurred in terms of the topological feature extraction._
> >
> > **A**: We provide an estimate of computational time using CodeLlama-7B model and HumanEval dataset as an example. The average time taken to generate a solution for one problem without feature extraction is 6,2 sec, with feature extraction 11,5 sec for all layers and heads, which are processed in parallel.
> > We also note that feature computation time during inference can be further decreased if only the most important features are further used for classification: according to Section 5.5, it is possible to use only 5% of features without significant decrease in performance.
> > The average memory footprint is ~190Mb for HumanEval and ~544Mb for MBPP. This size is a small fraction of GPU memory footprint during generation. We highlight that we extract the attention features on the fly during generation and do not store the heavyweight attention matrices.

---

### Official Review · Reviewer_CScZ · 2025-10-30

**Soundness:** 2
**Presentation:** 3
**Contribution:** 3
**Rating:** 4
**Confidence:** 4

**Summary:**

This paper addresses the automatic assessment of code correctness without using test cases. The proposed approach uses the attention maps of a code LLM to extract topological features, which in turn are used to train an xgboost classifier. The topological features come from the prior work of Barannikov et al. (2021) and Chazal & Michel (2017). The authors show that using these features outperforms many well-thought-out baselines.

**Strengths:**

This paper addresses the correctness assessment of code generated by LLMs. The addressed problem is important because correctness testing may be costly in time, and traditional methods require curation of test cases, where good coverage is challenging to achieve. Moreover, LLM-generated code may contain security vulnerabilities, so a prediction before code execution is desirable.

The authors have done a very comprehensive evaluation of the proposed method. The experiments include not just comparison with well-thought-out baselines but also feature analysis, different classification/ranking setups, evaluations on various languages, transferability, and an ablation study. The empirical evaluation is solid.

**Weaknesses:**

It is a mystery why attention maps and the topological features are effective for code correctness prediction. There are a few cited references (see the Introduction and Related Work sections of the paper) showing the usefulness of attention maps, but the evidence is just empirical. Moreover, there lack of justification on why the specific feature, Manifold Topology Divergence (MTD), is a good choice (except that it happens to apply to the current problem).

The ablation study (see Table 9 in the appendix) suggests that the MTD feature is very much comparable to the diagonal attention features, and combining the two does not yield significant improvement. As a consequence, one can be convinced that attention maps are helpful for the problem at hand, but one does not necessarily need to use the MTD features; the diagonal features are equally effective.

Figure 4: It is unclear why the shown distributions suggest that the corresponding features are discriminative. In fact, the distributions have significant overlaps.

**Questions:**

See the Weakness section.

---

> ### Author Response · Authors · 2025-11-17
> **Response to Reviewer CScZ**
>
> Thank you for your feedback. Below we respond to the questions one by one.
>
> **W1**. _It is a mystery why attention maps and the topological features are effective for code correctness prediction._
>
> **A**. Manifold Topology Divergence (MTD) when applied to attention graphs, measures the strength of connectivity between tokens of the generation and tokens of the prompt. Attention graphs are derived from attention maps, where nodes are tokens and weights are 1 - attention.
> Low values of MTD mean the high connectivity 1) between tokens of the generation and the prompt 2) between tokens inside the generation. Examples of MTD scores of some heads are shown in Fig. 4.
> Lack of generation-prompt attention means that representations of tokens from generation doesn’t depend on prompt. That is, LLM drifts away from the prompt during generation and hallucinates.
>
> **W2**. _The ablation study (see Table 9 in the appendix) suggests that the MTD feature is very much comparable to the diagonal attention features, and combining the two does not yield significant improvement._
>
> **A**. Our main argument in the ablation study is that two types of attention features complement each other. We provide experiments for smaller models in the same setting, see Table 7 in Appendix I. For smaller models the effect is more pronounced: for QwenCoder-1.5b, QwenCoder-3b removal of MTD features drops ROC-AUC by 3-4% and F1 by 2-6.5%. The removal of diagonal features for QwenCoder-3b improves metrics by ~1.5%.
> The ablation study of feature types in transferability from MBPP to HE is presented in Table 8 in Appendix I. Without MTD features, ROC-AUC drops up to 9%, while the best performance is achieved by combining both types of features. Therefore, we propose to account for both types of features in our approach.
>
> **W3**. _It is unclear why the shown distributions suggest that the corresponding features are discriminative. In fact, the distributions have significant overlaps._
>
> **A**. Some individual features are stronger than baselines (mean log. prob, fine-tuned CodeT5-base, prompt/generation length) and are transferable across programming languages (Table 14 in Appendix).
> In our method, we propose to combine features with a XGBoost classifier.

---

### Official Review · Reviewer_V2ph · 2025-11-01

**Soundness:** 2
**Presentation:** 2
**Contribution:** 2
**Rating:** 4
**Confidence:** 4

**Summary:**

This paper presents a new hallucination detection method for code generation. Their approach analyzes the attention maps using topological data analysis. The attention map is represented as a weighted graph, and its structure is quantified using Manifold Topology Divergence (MTD) metrics between the prompt and the generated code. These topological features are used to train an XGBoost classifier to predict whether generated code is correct without executing it. Experiments on HumanEval and MBPP across five code LLMs demonstrate higher ROC-AUC and F1-scores than other baselines.

**Strengths:**

1. The method uses the topology method to interpret and access the LLM attention map, which is innovative.
2. The method solves a meaningful task. It detects the correctness of generated code without executing it.
3. The experiment on five code LLMs and two benchmarks demonstrates strong performance, outperforming other baselines.

**Weaknesses:**

1. While this paper introduces a topology analysis method for hallucination detection, it lacks a formal argument or theoretical proof of the relationship between topological divergence and hallucination.
2. This paper provides insufficient details about how the training data is collected for the classifier. If the training data are derived from the test benchmarks, there is a potential risk of data leakage.
3. The derivation of the symmetrized weight w{ij} = 1 - \max(a{ij}, a_{ji}) (Line 193) is heuristic. Justification for this equation or alternative design is not discussed.
4. Some baselines (e.g., Self-Eval) are adapted from general-purpose LLMs. More advanced methods like CODEJUDGE (EMNLP 2024) are not discussed.
5. Since this paper is to determine the correctness of generated code, it would be valuable to compare with traditional program analysis methods (e.g., control flow graph analysis), which may be more lightweight.

**Questions:**

1. Though section 5.6 provide the classification fo error types, including failure case study would help strengthen the paper.
2. What is the advantage of this method compare traditional static analysis methods, such as control flow graph analysis?

---

> ### Author Response · Authors · 2025-11-21
> **Response to Reviewer V2ph, part 1/3**
>
> **W1**. _While this paper introduces a topology analysis method for hallucination detection, it lacks a formal argument or theoretical proof of the relationship between topological divergence and hallucination._
>
> **A**.  Manifold Topology Divergence (MTD), when applied to attention graphs, measures the strength of connectivity between tokens of the generation and tokens of the prompt. Attention graphs are derived from attention maps, where nodes are tokens and weights are 1 - attention. Low values of MTD mean the high connectivity 1) between tokens of the generation and the prompt 2) between tokens inside the generation. MTD scores of some heads have a significant discriminative power and are shown in Fig. 4.
> Lack of generation-prompt attention means that representations of tokens from generation doesn’t depend on prompt. That is, LLM drifts away from the prompt during generation and hallucinates.
>
> **W2**. _This paper provides insufficient details about how the training data is collected for the classifier. If the training data are derived from the test benchmarks, there is a potential risk of data leakage._
>
> **A**. We utilize two popular benchmarks for evaluation of code generation: HumanEval and MBPP. We obtain 25 generations per task for HumanEval and 5 generations per task for MBPP unless otherwise specified using temperature sampling with T=0.8. To address the quality of the proposed approach in different LLM prompting regimes, we used 0-shot prompt for the HumanEval dataset and 1-shot prompt for the MBPP dataset. We applied 5-fold stratified group cross-validation where different solutions of the same coding problem belonged to the same group. In this way, training and testing were performed always at non-overlapping coding problems (prompts). For each fold, all code generations for the same problem appear either in train or in test but not in both. The reported results are the mean and standard deviation estimated over the 5 folds. These details are covered in Section 5.1, 5.2 and Appendix A.
>
> **W3**. The derivation of the symmetrized weight w{ij} = 1 - \max(a{ij}, a_{ji}) (Line 193) is heuristic. Justification for this equation or alternative design is not discussed.
>
> **A**. LLMs in our study are decoder-only architectures having causal attention maps, where elements above the diagonal are zero.
> The symmetrization formula $w_{i,j} = 1 - \max(a_{i,j}, a_{j, i})$, for $i \neq j$ is technical. High attention means low “distance” between tokens.
> We are reformulating this paragraph to make it more clear.
>
> **W4**. Some baselines (e.g., Self-Eval) are adapted from general-purpose LLMs. More advanced methods like CODEJUDGE (EMNLP 2024) are not discussed.
>
> **A**. Thank you for your remark. We provide additional comparison for the CODEJUDGE baseline below. We use CodeLlama-Instruct 34B as an evaluation model to produce binary output whether the generated code is correct or not. We report the mean results over 3 runs in accordance with the original setup.
>
> | |StarCoder2-7b| |CodeLlama-7b| |DeepSeekCoder-6.7b| | QwenCoder-7b| | MagiCoder-6.7b| | CodeLlama-34b| | DeepSeek-Coder-33b| | Qwen2.5-Coder-32b| |
> |-|-|-|-|-|-|-|-|-|-|-|-|-|-|-|-|-|
> | |ROC-AUC|F1-score|ROC-AUC|F1-score|ROC-AUC|F1-score|ROC-AUC|F1-score|ROC-AUC|F1-score|ROC-AUC|F1-score|ROC-AUC|F1-score|ROC-AUC|F1-score|
> |CodeJudge A.S. w/o reference|70,55 ± 4,68|64,75 ± 4,32|71,32 ± 4,09|59,19 ± 3,77|69,16 ± 3,19|74,39 ± 4,76|67,24 ± 5,72|78,12 ± 3,46|55,32 ± 2,74|83,28 ± 6,60|65,26 ± 3,00|69,55 ± 4,80|72,01 ± 7,2|80,22 ± 3,84|61,61 ± 7,26|81,83 ± 4,69|
> |CodeJudge A.S. w/ reference|75,76 ± 4,71|68,23 ± 3,91|72,98 ± 3,31|61,02 ± 5,30|68,78 ± 7,92|70,92 ± 8,99|70,38 ± 7,40|77,59 ± 4,30|57,93 ± 7,14|82,08 ± 3,55|68,72 ± 3,79|70,01 ± 5,36|75,83 ± 5,60|81,05 ± 2,37|61,99 ± 7,49|80,26 ± 1,95|
> |Attn. Feat.|80,8 ± 3,7|63,5 ± 9,1|80,1 ± 3,9|59,9 ± 9,4|83,9 ± 8,4|78,5 ± 6,7|73,3 ± 7,1|73,7 ± 6,1|65,1 ± 3,4|85,4 ± 5,4|75,0 ± 5,1|68,4 ± 5,9|86,5 ± 7,2|81,6 ± 4,0|74,0 ± 6,4|82,7 ± 1,6|
>
> We highlight that for fair comparison we should consider the setup when CODEJUDGE is run without reference and Attn. Feat. is run for large 32-34b models. In our work, we did not include reference (i.e. correct solution) to the prompt as this setup is not practical (typically, one does not know the correct solution). Also, since CODEJUDGE is evaluated with a 34b model, we evaluate the Attn. Feat. in a similar way to keep the experimental design of both methods as close as possible. In this comparison, Attn. Feat. outperforms CODEJUDGE in the majority of cases, winning by a large margin for ROC-AUC. However, to further explore the capabilities of the proposed Attn. Feat., we provide additional comparison when CODEJUDGE is run with reference and when Attn. Feat. is run for smaller 6.7-7b models. In these cases, Attn. Feat. still outperforms CODEJUDGE as measured by ROC-AUC although the F1-score performance is lower in some cases.

---

> > ### Author Response · Authors · 2025-11-21
> > **Response to Reviewer V2ph, part 2/3**
> >
> > **W5**. _Since this paper is to determine the correctness of generated code, it would be valuable to compare with traditional program analysis methods (e.g., control flow graph analysis), which may be more lightweight._
> >
> > **A**. We provide additional comparison with Pylint, a static code analysis tool for Python, see Table below. In this experiment, we run Pylint on the code generations from each of the code LLM and parse the result. We treat outputs with “Error” and “Fatal” marks as incorrect while other cases are considered correct. For fair comparison, we split the data similarly to our main results (i.e. w.r.t. tasks) and measure ROC-AUC and F1-score of parsed Pylint outputs as predictions. In the majority of cases, the proposed classifier based on attention features outperforms the Pylint detection by a large margin.
> >
> > | | |HumanEval| | MBPP| |
> > |-|-|-|-|-|-|
> > | | |ROC-AUC |F1-score|ROC-AUC|F1-score|
> > | starcoder2-7b | Pylint | 58.9 ± 1.0 | 49.6 ± 4.7 | 53.0 ± 0.6 | 61.4 ± 1.0 |
> > |               | Attn. feat. | 82.9 ± 2.7 | 54.2 ± 6.9 | 81.9 ± 2.4 | 68.4 ± 5.3 |
> > | CodeLlama-7B  | Pylint | 55.1 ± 0.3 | 43.7 ± 4.8 | 53.1 ± 0.5 | 53.5 ± 1.1 |
> > |               | Attn. feat. | 85.6 ± 3.9 | 56.4 ± 7.2 | 83.4 ± 3.3 | 64.0 ± 4.4 |
> > | deepseek-coder-6.7b-base | Pylint | 54.6 ± 0.8 | 59.7 ± 3.1 | 52.8 ± 0.4 | 70.1 ± 1.6 |
> > |               | Attn. feat. | 85.6 ± 2.8 | 68.9 ± 5.5 | 82.6 ± 1.9 | 76.5 ± 2.7 |
> > | Qwen2.5-Coder-7B | Pylint | 64.1 ± 2.2 | 71.4 ± 6.3 | 62.0 ± 2.5 | 74.1 ± 0.8 |
> > |               | Attn. feat. | 81.7 ± 2.8 | 70.2 ± 4.2 | 82.2 ± 2.2 | 75.4 ± 1.7 |
> > | Magicoder-S-DS-6.7B | Pylint | 52.7 ± 1.3 | 80.0 ± 3.3 | 52.0 ± 0.8 | 76.7 ± 2.4 |
> > |               | Attn. feat. | 82.3 ± 4.9 | 80.7 ± 3.6 | 81.5 ± 1.6 | 80.6 ± 2.0 |
> >
> > **Q1**. _Though section 5.6 provides the classification of error types, including failure case study would help strengthen the paper._
> >
> > **A**: We provide a failure case study in the table below. For each trained classifier, we analyze the category of generated code which was classified incorrectly by the classifier. We report the fraction of top-3 most popular code categories w.r.t. number of samples in the test sample. We use 5-fold stratified group cross validation and report the average values over 5 folds. The table reveals the most popular failure cases: misclassification of correct codes that passed the tests (referred to as ‘passed’) and misclassification of wrong solutions (i.e. codes that did not pass any test and caused an AssertionError). The fraction of other errors (top-3 and others) is sufficiently lower.
> >
> > | |HumanEval| | | MBPP| | |
> > |-|-|-|-|-|-|-|
> > | | Top-1| Top-2| Top-3| Top-1| Top-2| Top-3|
> > |StarCoder2-7b| Passed 17,16 ± 4,49| AssertionError 2,93 ± 2,23| NameError 0,75 ± 0,45| Passed 15,0 ± 3,46| AssertionError 9,68 ± 2,21| NameError 0,68 ± 0,37|
> > |CodeLLama-7b| Passed 14.6 ± 3.6 | AssertionError 3.8 ± 2.7 | NameError 0.2 ± 0.2 |Passed 15.5 ± 1.6|AssertionError 7.4 ± 1.2 | NameError 0.5 ± 0.4 |
> > |DeepSeekCoder-6.7b|Passed 14,56 ± 2,85|AssertionError 4,89 ± 1,15|IndexError 0,49 ± 0,57|AssertionError 12,64 ± 3,19|Passed 11,08 ± 3,77|NameError 0,68 ± 0,48|
> > |QwenCoder-7b|Passed 15,96 ± 5,39|AssertionError 6,93 ± 2,77|NameError 2,09 ± 0,72|Passed 11,32 ± 0,5|AssertionError 10,8 ± 1,36|NameError 1,32 ± 0,45|
> > |Magicoder-6.7b|AssertionError 12,11 ± 4,83|Passed 10,4 ± 4,82|IndexError 0,71 ± 0,57|AssertionError 14,52 ± 4,42|Passed 9,24 ± 2,3|TypeError 0,56 ± 0,5|

---

> > > ### Author Response · Authors · 2025-11-21
> > > **Response to Reviewer V2ph, part 3/3**
> > >
> > > **Q2**. _What is the advantage of this method compare traditional static analysis methods, such as control flow graph analysis?_
> > >
> > > **A**. In the Table below we provide analysis of error distribution. We focus on two types of errors: assertion error and errors of other types. Assertion errors happen when the generated code fails to pass any of the given tests which indicates the incorrect logic of the candidate solution. Other errors include all other types of errors appearing in generated code (e.g. AttributeError, IndentationError, IndexError, NameError, SyntaxError, etc.). We suppose they are less related to the logic of the solution but rather represent the ability to generate runnable/compiled code. The table below highlights that even in 7b-models the assertion errors strongly dominate all other errors. Traditional static analysis methods can’t identify assertion errors by construction. On the contrary, the proposed approach is developed to utilize the structure of Code LLM internal knowledge (provided by attention scores), in particular, topological alignment between prompt and generation, to complement other analysis tools to identify incorrect logic in generated solutions. This result is confirmed in the Table from the previous question.
> > > | | Assertion fraction, % | Other errors fraction, % |
> > > |-|-|-|
> > > | starcoder2-7b |
> > > | humaneval | 71.5 | 28.5 |
> > > | mbpp | 85.1 | 14.9 |
> > > | CodeLlama-7B |  | |
> > > | humaneval | 79.3 | 20.7 |
> > > | mbpp | 83.4 | 16.5 |
> > > | deepseek-coder-6.7b-base | |
> > > | humaneval | 81.7 | 18.3 |
> > > | mbpp | 86.3 | 13.4 |
> > > | Qwen2.5-Coder-7B | |
> > > | humaneval | 62.9 | 37.1 |
> > > | mbpp | 66.9 | 33.1 |
> > > | Magicoder-S-DS-6.7B |
> > > | humaneval | 83.1 | 16.9 |
> > > | mbpp | 90.6 | 9.4 |

---

> ### Comment · Reviewer_V2ph · 2025-11-26
> **Response to the rebuttal**
>
> Thank you for the responses. I appreciate the clarification in W1, but the causal link is still unclear. You state that "lack of generation-prompt attention means the model drifts away," but attention drift does not necessarily imply hallucination. The model may simply be generating correct continuation code that pays little attention to corresponding tokens. Could you provide a concrete visualization or empirical example showing how MTD captures attention drift and how this drift correlates with actual hallucination?
>
> Regarding W3, describing the symmetrization choice as merely "technical" is not sufficient. Because decoder-only models have causal masking, choosing max or sum can alter the resulting graph topology and thus impact MTD. Did you run any ablation study showing that max empirically outperforms these alternatives?
>
> For Q1, thank you for providing the distribution of error types. However, to fully understand where the topological method fails, it would be helpful to include sa pecific example.

---

> > ### Author Response · Authors · 2025-11-27
> > **Response from Authors, part 1/2**
> >
> > **Q1**. _On symmetrization of attention maps_.
> >
> > **A**. In causal attention $a_{i,j} = 0$ for $j > i$.
> > Thus $w_{i,j} = 1 - \max(a_{i,j}, a_{j, i}) = 1 - (a_{i,j} + a_{j, i})$ so max = sum and ablation is irrelevant.
> > We have updated our manuscript to make it more clear: Line 192-193.
> >
> > **Q2**. _On specific examples of misclassification_.
> >
> > **A**.  We took pieces of code from CodeLLama-7B, HumanEval benchmark, which were wrongly classified and used DeepSeek to summarize patterns where our classifier fails.
> > We used a random sample of 50 examples of two most frequent misclassification types. Results are below.
> >
> > A) False negatives: correct code, classified as a "hallucination":
> >
> > 1. Algorithmic Complexity & Efficiency Issues (25%). Misjudging time/space complexity of valid solutions.
> > 2. Edge Case Handling (20%). Overlooking proper handling of empty inputs, zero values, and boundary conditions.
> > 3. Alternative Implementation Approaches (18%). Expecting specific algorithms when multiple valid approaches exist.
> > 4. Code Style & Structure Variations (17%). Penalizing different but valid control flow structures.
> > 5. Mathematical & Logical Nuances (20%). Missing mathematical equivalencies and transformations.
> >
> > The ML classifier appears to struggle most with recognizing that multiple valid approaches can solve the same problem, particularly when implementations differ from "textbook" examples or use practical optimizations that maintain correctness while potentially sacrificing theoretical purity.
> >
> > B) False positives: syntactically correct code which failed functional tests, classified as "correct".
> >
> > 1. Logical Errors in Algorithm Implementation (40%). The code follows the right general approach but contains flaws in core logic.
> > 2. Edge Case Handling Failures (25%). The code works for typical inputs but fails on edge cases.
> > 3. Incomplete or Incorrect Problem Understanding (15%). The solution addresses part of the requirement but misses key aspects.
> > 4. Data Structure Misuse (12%). Using inappropriate data structures for the task.
> > 5. State Management Errors (8%). Problems with tracking program state across iterations
> >
> > The ML classifier appears to be most vulnerable to subtle logical flaws that don't manifest as obvious syntax or runtime errors, particularly those that require deeper semantic understanding of the problem domain and edge case analysis.
> >
> > **Q3**. _On drifting away from the prompt during generation_.
> >
> > **A**. We show below code samples from Code LLMs having top values of the most important MTD features. These are the cases where Code LLMs “drift away from the prompt”. Code LLM doesn't generate code et all, misses instructions in prompt, or loops in repetitions.
> >
> > **StarCoder2-7b**, Top performing feature – MTD_0(G, P) / |P|
> >
> > ```python
> > def car_race_collision(n: int):
> >     """
> >     Imagine a road that's a perfectly straight infinitely long line.
> >     n cars are driving left to right;  simultaneously, a different set of n cars
> >     are driving right to left.   The two sets of cars start out being very far from
> >     each other.  All cars move in the same speed.  Two cars are said to collide
> >     when a car that's moving left to right hits a car that's moving right to left.
> >     However, the cars are infinitely sturdy and strong; as a result, they continue moving
> >     in their trajectory as if they did not collide.
> >
> >     This function outputs the number of such collisions.
> >     """
> >     collisions = 0
> >     for i in range(n):
> >         collisions += 1  # one collision is possible for every car pair
> >     return collisions
> > ```
> > ```python
> > def car_race_collision(n: int):
> >     """
> >     Imagine a road that's a perfectly straight infinitely long line.
> >     n cars are driving left to right;  simultaneously, a different set of n cars
> >     are driving right to left.   The two sets of cars start out being very far from
> >     each other.  All cars move in the same speed.  Two cars are said to collide
> >     when a car that's moving left to right hits a car that's moving right to left.
> >     However, the cars are infinitely sturdy and strong; as a result, they continue moving
> >     in their trajectory as if they did not collide.
> >
> >     This function outputs the number of such collisions.
> >     """
> >     # YOUR CODE HERE
> >     raise NotImplementedError()
> > ```

---

> ### Author Response · Authors · 2025-11-27
> **Response from Authors, part 2/2**
>
> **CodeLLama-7B**, MTD_0(P,G) from layer 1, head 7 having a high discriminative power (Figure 4, left).
> ```python
> def car_race_collision(n: int):
>     """
>     Imagine a road that's a perfectly straight infinitely long line.
>     n cars are driving left to right;  simultaneously, a different set of n cars
>     are driving right to left.   The two sets of cars start out being very far from
>     each other.  All cars move in the same speed.  Two cars are said to collide
>     when a car that's moving left to right hits a car that's moving right to left.
>     However, the cars are infinitely sturdy and strong; as a result, they continue moving
>     in their trajectory as if they did not collide.
>
>     This function outputs the number of such collisions.
>     """
>
>     pass
> ````
>
> ```python
> def Strongest_Extension(class_name, extensions):
>     """You will be given the name of a class (a string) and a list of extensions.
>     The extensions are to be used to load additional classes to the class. The
>     strength of the extension is as follows: Let CAP be the number of the uppercase
>     letters in the extension's name, and let SM be the number of lowercase letters
>     in the extension's name, the strength is given by the fraction CAP - SM.
>     You should find the strongest extension and return a string in this
>     format: ClassName.StrongestExtensionName.
>     If there are two or more extensions with the same strength, you should
>     choose the one that comes first in the list.
>     For example, if you are given "Slices" as the class and a list of the
>     extensions: ['SErviNGSliCes', 'Cheese', 'StuFfed'] then you should
>     return 'Slices.SErviNGSliCes' since 'SErviNGSliCes' is the strongest extension
>     (its strength is -1).
>     Example:
>     for Strongest_Extension('my_class', ['AA', 'Be', 'CC']) == 'my_class.AA'
>     """
>     # Strength of the extension is as follows:
>     # Let CAP be the number of the uppercase letters in the extension's name,
>     # and let SM be the number of lowercase letters in the extension's name,
>     # the strength is given by the fraction CAP - SM.
>
>     # You should find the strongest extension and return a string in this
>     # format: ClassName.StrongestExtensionName.
>     # If there are two or more extensions with the same strength, you should
>     # choose the one that comes first in the list.
>     # For example, if you are given "Slices" as the class and a list of the
>     # extensions: ['SErviNGSliCes', 'Cheese', 'StuFfed'] then you should
>     # return 'Slices.SErviNGSliCes' since 'SErviNGSliCes' is
> ```
>
> **Qwen2.5-Coder-7B**, Top performing feature – MTD_0 (P, G) / |G|
>
> ```
> def is_sorted(lst):
>     '''
>     Given a list of numbers, return whether or not they are sorted
>     in ascending order. If list has more than 1 duplicate of the same
>     number, return False. Assume no negative numbers and only integers.
>
>     Examples
>     is_sorted([5]) ➞ True
>     is_sorted([1, 2, 3, 4, 5]) ➞ True
>     is_sorted([1, 3, 2, 4, 5]) ➞ False
>     is_sorted([1, 2, 3, 4, 5, 6]) ➞ True
>     is_sorted([1, 2, 3, 4, 5, 6, 7]) ➞ True
>     is_sorted([1, 3, 2, 4, 5, 6, 7]) ➞ False
>     is_sorted([1, 2, 2, 3, 3, 4]) ➞ True
>     is_sorted([1, 2, 2, 2, 3, 4]) ➞ False
>     '''
>     return sorted(lst) == lst
> ```
> ```python
> def is_nested(string):
>     '''
>     Create a function that takes a string as input which contains only square brackets.
>     The function should return True if and only if there is a valid subsequence of brackets
>     where at least one bracket in the subsequence is nested.
>
>     is_nested('[[]]') ➞ True
>     is_nested('[]]]]]]][[[[[]') ➞ False
>     is_nested('[][]') ➞ False
>     is_nested('[]') ➞ False
>     is_nested('[[][]]') ➞ True
>     is_nested('[[]][[') ➞ True
>     '''
>     brackets = []
>     for char in string:
>         if char == '[':
>             brackets.append(char)
>         elif char == ']':
>             if len(brackets) > 0 and brackets[-1] == '[':
>                 # brackets.pop()
>                 # if brackets:
>                 #     brackets.pop()
>                 # else:
>                 #     return False
>                 # if len(brackets) > 0:
>                 #     brackets.pop()
>                 # else:
>                 #     return False
>                 # if brackets:
>                 #     brackets.pop()
>                 # else:
>                 #     return False
>                 # if brackets:
>                 #     brackets.pop()
>                 # else:
>                 #     return False
>                 # if brackets:
> ```

---

### Official Review · Reviewer_FxPf · 2025-11-01

**Soundness:** 2
**Presentation:** 2
**Contribution:** 2
**Rating:** 4
**Confidence:** 3

**Summary:**

The authors propose a method to detect "hallucinations" in code LLMs before executing the code. The key idea is to treat the attention matrices between prompts and generated code as weighted complete graphs, then compute topological features like Cross-Barcode/MTD based on Topological Data Analysis (TDA), and use a lightweight classifier to predict correctness. The method outperforms various baselines on HumanEval and MBPP across multiple models (StarCoder2-7B, CodeLlama-7B, DeepSeek-Coder-6.7B, Qwen2.5-Coder-7B, Magicoder-S-DS-6.7B), and the trained classifiers can transfer between the two benchmarks. They symmetrize the attention matrix ($A=[a_{ij}]$) into edge weights ($w_{ij}=1-max(a_{ij},a_{ji}) for i≠j$), extract barcodes from the filtration as thresholds increase, and aggregate them into vertex-set-normalized features (MTD₀, MTD₁) plus diagonal attention sums. These features are computed layer-by-layer and head-by-head, then fed into an XGBoost binary classifier. The full pipeline goes: "concatenate prompt + generation, extract attention, build graph, compute Cross-Barcode, train classifier".
Experimental results show that these topological features effectively discriminate between hallucinated and correct code distributions. Interestingly, only a small subset of heads/layers is needed to approach full performance, suggesting the relevant information is concentrated in a few attention heads. The authors note that future work will focus on pinpointing the specific buggy locations in the code.

**Strengths:**

# Strengths

This paper proposes a novel approach to detecting hallucinations in code LLMs by applying Topological Data Analysis (TDA) to attention maps. The core innovation lies in representing attention maps as weighted graphs and computing Manifold Topology Divergence (MTD) features to quantify the dissimilarity between prompt and generated code token interactions, providing an interpretable geometric perspective on model failures. The technical execution is rigorous, incorporating thoughtful methodological choices: symmetrizing attention maps into undirected graphs, extracting H₀ (connected components) and H₁ (cycles) topological features, and including diagonal attention values for completeness. The authors also find that only approximately 5% of attention features are needed to achieve high performance, suggesting that specific attention heads can serve as reliable indicators of code correctness. The visualization design (Figures 1-2 illustrating MTD intuition, Figures 11-14 showing cross-barcodes and attention patterns) effectively helps readers understand the topological approach, and the paper is well-structured with comprehensive appendices supporting reproducibility.

The experimental evaluation is thorough and demonstrates strong empirical results across multiple dimensions. The authors tested 5 different code LLMs (with parameters ranging from 6.7B to 7B), two benchmarks (HumanEval and MBPP), and multiple programming languages (Python, Java, Go, Rust, Lua), achieving consistent ROC-AUC improvements of 10-20% over baseline methods. The experimental design employs proper 5-fold stratified group cross-validation to prevent data leakage between training and testing sets, and includes valuable transferability analysis (Table 4), feature importance analysis (Section 5.5), multi-class error detection (Table 5), and practical ranking applications (Table 3 showing 10-15% pass@1 improvements).

**Weaknesses:**

1. While the authors provide an intuitive explanation that "attention drifting away from the prompt" causes hallucinations, this only explains one failure mode. Section 5.6 shows that the method successfully detects multiple error types (SyntaxError, ZeroDivisionError, NameError, etc.), yet the paper fails to explain how these different failures manifest in attention topology.

2. The symmetrization formula $w_{i,j} = 1 - max(a_{i,j}, a_{j,i})$ lacks theoretical justification, and there are no ablation experiments comparing alternative graph construction methods (thresholding, asymmetric graphs, different aggregation functions).

3. More critically, there is a lack of causal analysis: do incorrect codes inherently possess different attention topology, or is this correlation driven by confounding factors (such as code length or complexity)?

4. The transferability results (Table 4) reveal concerning performance degradation. For example, CodeLlama's ROC-AUC drops from 85.6 to 69.5 when transferring from HE→MBPP, and the best transfer performance requires combining attention features with mean log probability, indicating that attention features alone are insufficient. The paper does not analyze the distribution shift factors causing transfer failures (prompt structure? code style? problem difficulty?), includes no cross-model transferability experiments (can a classifier trained on StarCoder2 be applied to CodeLlama?), and only tests a single temperature parameter (T=0.8), whereas production systems typically use greedy decoding.

5. The experimental design and baseline comparisons have significant limitations that restrict the scope of conclusions. The baseline methods do not include any static analysis tools (CodeQL, Pylint, basic syntax checkers) that can capture many syntax and some logic errors, nor do they include execution-based methods (such as running partial test suites).

6. Practical deployment faces challenges: computing MTD for all heads and layers incurs non-trivial overhead (no timing analysis is provided), the method requires storing attention maps (memory issues are acknowledged but not addressed), and it needs labeled training data, which is impractical for new domains without existing functional tests.

**Questions:**

See the previous section

---

> ### Author Response · Authors · 2025-11-19
> **Response to Reviewer FxPf, part 1/2**
>
> **W1**. _Section 5.6 shows that the method successfully detects multiple error types (SyntaxError, ZeroDivisionError, NameError, etc.), yet the paper fails to explain how these different failures manifest in attention topology._
>
> **A**. Some features can distinguish AssertionError (functional test failed) vs. other errors (SyntaxError, ZeroDivisionError, NameError, etc.). Here are the results for CodeLLama-7B, HumanEval dataset:
>
> * answer diag.attention, layer 29, head 24:  ROC_AUC = 0.67
> * MTD$_0$(P, G) layer 0, head 4:  ROC_AUC = 0.66
> * MTD$_0$(P, G) layer 1, head 10: ROC_AUC = 0.66
>
> We leave a more detailed study of how different error types manifest in attention topology to further work. Figures 13, 14 in Appendix shows attention maps for correct/incorrect generations, Figure 12 in Appendix shows examples of Cross-Barcode$_0$(G, P), Cross-Barcode$_1$(G, P) for  CodeLLama-7B, HumanEval dataset and a fixed attention head.
>
> **W2**. _The symmetrization formula lacks theoretical justification, and there are no ablation experiments comparing alternative graph construction methods (thresholding, asymmetric graphs, different aggregation functions)._
>
> **A**. LLMs in our study have a decoder-only architecture and lower triangular attention maps, where elements above the diagonal are zero.
> The symmetrization formula $w_{i,j} = 1 - \max(a_{i,j}, a_{j, i})$, for $i \neq j$ is technical.
> Tools of persistent homology, MTD in particular, provide an opportunity to analyze graph structure for all thresholds of $w_{i,j}$  at once.
> We are reformulating this paragraph to make it more clear.
>
> **W3**. _More critically, there is a lack of causal analysis: do incorrect codes inherently possess different attention topology, or is this correlation driven by confounding factors (such as code length or complexity)?_
>
> **A**. The code length has a negative correlation with code correctness (Fig. 8, 9 in Appendix). We use prompt and generation lengths as baselines and empirically show that our method outperforms them (Table 2) by a large margin. All topological and attention features are length-normalized (Section 4). Thus, the performance of topological attention features are not driven by code length.
>
> **W4.1**. _The transferability results (Table 4) reveal concerning performance degradation. The paper does not analyze the distribution shift factors causing transfer failures._
>
> **A**. We consider that the drop when transferring HE→MBPP is due to a different prompt structure (0-shot for HE, 1-shot for MBPP).
>
> **W4.2**. _Greedy decoding is not studied_.
>
> **A**. Results for greedy decoding are below. Our method shows consistently high performance.
>
> | |HumanEval | | MBPP | |
> |-|-|-|-|-|
> | | _ROC-AUC_ | _F1-score_ | _ROC-AUC_ | _F1-score_ |
> | StarCoder2-7B | 80,8 ± 3,7 | 61,9 ± 9,5 | 79,3 ± 6,1 |  69,7 ± 5,4 |
> | CodeLlama-7B | 80,1 ± 3,9 | 59,4 ± 10,0 | 82,3 ± 3,1 | 71,3 ± 6,3 |
> | DeepSeek-Coder-6.7B | 83,9 ± 8,4 | 78,5 ± 6,7 | 79,6 ± 3,4 | 78,7 ± 2,1 |
> | Qwen2.5-Coder-7B | 73,3 ± 7,1 | 70,1 ± 6,4 | 74,0 ± 1,9 | 75,7 ± 4,0 |
> | Magicoder-S-DS-6.7B | 65,1 ± 3,4 | 82,6 ± 4,3 | 79,3 ± 2,0 | 80,8 ± 1,3 |
>
> **W4.3**. _On cross-model transferability experiments: can a classifier trained on StarCoder2 be applied to CodeLlama?_
>
> **A**. Results for cross-model transferability are below. We conclude that a classifier trained on DeepSeek-Coder-6.7B is transferable to Magicoder-S-DS-6.7B and vice versa. But it is not the case for CodeLlama-7B. The reason is that Magicoder-S-DS-6.7B is a fine-tuned DeepSeek-Coder-6.7B and a correspondence between attention heads persists.
>
> **HumanEval**
> | Test | CodeLlama-7B | | DeepSeek-Coder-6.7B | | Magicoder-S-DS-6.7B | |
> |-|-|-|-|-|-|-|
> | **Train** | _ROC-AUC_ | _F1-score_ |  _ROC-AUC_ | _F1-score_ | _ROC-AUC_ |  _F1-score_ |
> | CodeLlama-7B | 85,6 ± 3,9 | 56,4 ± 7,2 | 51,9 ± 13,2 | 17,3 ± 21,1 | 51,1 ± 7,1 | 24,3 ± 29,0 |
> | DeepSeek-Coder-6.7B | 45,4 ± 10,3 | 14,2 ± 9,6 | 85,6 ± 2,8 | 68,9 ± 5,5 | 79,4 ± 3,7 | 69,4 ± 6,8 |
> | Magicoder-S-DS-6.7B | 51,8 ± 5,4 | 23,4 ± 17,7 | 83,0 ± 2,2 | 70,7 ± 2,7 | 82,3 ± 4,9 | 80,7 ± 3,6 |
>
> **MBPP**
> | Test | CodeLlama-7B | | DeepSeek-Coder-6.7B | | Magicoder-S-DS-6.7B | |
> |-|-|-|-|-|-|-|
> | **Train** | _ROC-AUC_ | _F1-score_ | _ROC-AUC_ | _F1-score_ | _ROC-AUC_ | _F1-score_ |
> | CodeLlama-7B | 83.4 ± 3.3 | 64.0 ± 4.4 | 49.3 ± 2.9 | 17.2 ± 22.6 | 52.3 ± 9.4 | 21.9 ± 26.8 |
> | DeepSeek-Coder-6.7B | 49.2 ± 5.5 | 21.0 ± 17.8 | 82.6 ± 1.9 | 76.5 ± 2.7 | 75.2 ± 1.6 | 54.5 ± 11.3 |
> | Magicoder-S-DS-6.7B | 51.8 ± 3.4 | 29.2 ± 24.0 | 64.8 ± 6.4 | 58.5 ± 13.9 | 81.5 ± 1.6 | 80.6 ± 2.0 |

---

> > ### Author Response · Authors · 2025-11-19
> > **Response to Reviewer FxPf, part 2/2**
> >
> > **W5**. _The baseline methods do not include any static analysis tools (CodeQL, Pylint, basic syntax checkers) that can capture many syntax and some logic errors, nor do they include execution-based methods (such as running partial test suites)._
> >
> > **A**. We provide additional comparison with Pylint, a static code analysis tool for Python, see Table below. In this experiment, we run Pylint on the code generations from each of the code LLM and parse the result. We treat outputs with “Error” and “Fatal” marks as incorrect while other cases are considered correct. For fair comparison, we split the data similarly to our main results (i.e. stratified by tasks) and measure ROC-AUC and F1-score of parsed Pylint outputs as predictions. In the majority of cases, the proposed classifier based on attention features outperforms the Pylint detection by a large margin.
> >
> > | | |HumanEval| | MBPP| |
> > |-|-|-|-|-|-|
> > | | |ROC-AUC |F1-score|ROC-AUC|F1-score|
> > | starcoder2-7b | Pylint | 58.9 ± 1.0 | 49.6 ± 4.7 | 53.0 ± 0.6 | 61.4 ± 1.0 |
> > |               | Attn. feat. | **82.9 ± 2.7** | **54.2 ± 6.9** | **81.9 ± 2.4** | **68.4 ± 5.3** |
> > | CodeLlama-7B  | Pylint | 55.1 ± 0.3 | 43.7 ± 4.8 | 53.1 ± 0.5 | 53.5 ± 1.1 |
> > |               | Attn. feat. | **85.6 ± 3.9** | **56.4 ± 7.2** | **83.4 ± 3.3** | **64.0 ± 4.4** |
> > | deepseek-coder-6.7b-base | Pylint | 54.6 ± 0.8 | 59.7 ± 3.1 | 52.8 ± 0.4 | 70.1 ± 1.6 |
> > |               | Attn. feat. | **85.6 ± 2.8** | **68.9 ± 5.5** | **82.6 ± 1.9** | **76.5 ± 2.7** |
> > | Qwen2.5-Coder-7B | Pylint | 64.1 ± 2.2 | **71.4 ± 6.3** | 62.0 ± 2.5 | 74.1 ± 0.8 |
> > |               | Attn. feat. | **81.7 ± 2.8** | 70.2 ± 4.2 | **82.2 ± 2.2** | **75.4 ± 1.7** |
> > | Magicoder-S-DS-6.7B | Pylint | 52.7 ± 1.3 | 80.0 ± 3.3 | 52.0 ± 0.8 | 76.7 ± 2.4 |
> > |               | Attn. feat. | **82.3 ± 4.9** | **80.7 ± 3.6** | **81.5 ± 1.6** | **80.6 ± 2.0** |
> >
> >
> > The reason for Pylint's low performance is that code generated by modern Code LLMs has few syntactic errors. In the Table below we provide analysis of error distribution. Assertion errors happen when the generated code fails to pass any of the given functional tests. Other errors include all other types of errors appearing in generated code (e.g. AttributeError, IndentationError, IndexError, NameError, SyntaxError, etc.).
> >
> > | | Assertion fraction, % | Other errors fraction, % |
> > |-|-|-|
> > | starcoder2-7b |
> > | humaneval | 71.5 | 28.5 |
> > | mbpp | 85.1 | 14.9 |
> > | CodeLlama-7B |  | |
> > | humaneval | 79.3 | 20.7 |
> > | mbpp | 83.4 | 16.5 |
> > | deepseek-coder-6.7b-base | |
> > | humaneval | 81.7 | 18.3 |
> > | mbpp | 86.3 | 13.4 |
> > | Qwen2.5-Coder-7B | |
> > | humaneval | 62.9 | 37.1 |
> > | mbpp | 66.9 | 33.1 |
> > | Magicoder-S-DS-6.7B |
> > | humaneval | 83.1 | 16.9 |
> > | mbpp | 90.6 | 9.4 |
> >
> > It is not clear what Reviewer means by “running partial test suites”. The program is considered correct if it passes all the tests.
> >
> > **W6**. _Practical deployment faces challenges: computing MTD for all heads and layers incurs non-trivial overhead (no timing analysis is provided), the method requires storing attention maps (memory issues are acknowledged but not addressed), and it needs labeled training data, which is impractical for new domains without existing functional tests._
> >
> > **A**: We provide an estimate of computational time using CodeLlama-7B model and HumanEval dataset as an example. The average time taken to generate a solution for one problem without feature extraction is 6,2 sec, with feature extraction 11,5 sec for all layers and heads, which are processed in parallel.
> > Feature computation time during inference can be further decreased if only the most important features are used for classification: according to Section 5.5, it is possible to use only 5% of all attention features without significant loss of classification quality
> > The average memory footprint is ~190Mb for HumanEval and ~544Mb for MBPP. This size is a small fraction of GPU footprint during generation.
> > The classifiers are transferable to new domains, see Table 4.

---

### Author Response · Authors · 2025-11-24
**Summary of changes in the rebuttal revision**

We thank the Reviewers for their valuable feedback and suggested improvements to our paper. We addressed the questions raised by each Reviewer and uploaded the rebuttal revision, where we integrated the answers to the key questions and provided additional comments when necessary (see the rebuttal revision). For better visualization, the changes were highlighted with yellow. In the following, we summarize the major points.

- A formal argument behind the relationship between hallucination and attention topology (Reviewers V2ph, CScZ)

As mentioned in appendix H, in this work, we consider only one possible reason behind hallucination - when the generated code deviates from the problem formulation (prompt) reflected in the geometry of the model’s attention. Manifold Topology Divergence (MTD), when applied to attention graphs, measures the strength of connectivity between tokens of the generation and tokens of the prompt. Attention graphs are derived from attention maps, where nodes are tokens and weights are 1 - attention. Low values of MTD mean high connectivity 1) between tokens of the generation and the prompt 2) between tokens inside the generation. The lack of generation-prompt attention means that the representations of tokens from generation do not depend on prompt. That is, LLM drifts away from the prompt during generation and hallucinates. We add this clarification in Section 4.

- Justification for symmetrization formula (Reviewers FxPf, V2ph)

We highlight that LLMs in our study have a decoder-only architecture and lower triangular attention maps, where elements above the diagonal are zero. Thus, the symmetrization formula is a technical moment necessary to satisfy the requirements for MTD computation.

- Failure case study (Reviewer V2ph)

We add the failure case study in Appendix N. There are two most popular failure cases: misclassification of correct code that passed the tests and misclassification of wrong solutions. The fractions of other errors are sufficiently lower.

- Reasons for benchmark transfer failures (Reviewer FxPf)

Our motivation behind this experiment was to verify transferability between classifiers corresponding to different types of prompt used: 0-shot for HumanEval and 1-shot for MBPP. We attribute the performance differences to exactly this reason.

- Cross-model transferability (Reviewer FxPf)

We provide cross-model transferability results in Appendix M. We observe high transferability between associated DeepSeek-Coder-6.7b and Magicoder-S-DS-6.7b and sufficiently lower transferability with unrelated CodeLlama-7b.

- Evaluation with Greedy Decoding (Reviewer FxPf)

In Appendix L, we verify the proposed approach in additional settings when greedy decoding is used to generate train and test data. Although the number of labeled samples is small, the overall classifier performance is high. This also confirms that there is no need in large labeled samples to obtain robust performance.

- Additional comparison with static analysis tools, advantages of our approach (Reviewers FxPf, V2ph)

In Table 2 in the main part, we provide additional comparison with Pylint, a static code analysis tool for Python. In the majority of cases, the proposed classifier based on attention features outperforms Pylint detection by a large margin. We also highlight that while most errors in LLM generated code are caused by AssertionError, traditional static analysis methods can not identify Assertion Errors by construction. In contrast, as confirmed by experiments, our approach outperforms Pylint and is more universal.

- Additional comparison with CODEJUDGE (Reviewer V2ph)

We add a reference to CODEJUDGE to the Related Work section and provide a comparison in Appendix J. In fair comparison (similar model size and no solution provided), Attn. Feat. outperforms CODEJUDGE in the majority of cases, winning by a large margin for ROC-AUC. In other cases (smaller models for the Attn. Feat. classifier and provided a solution to CODEJUDGE), Attn. Feat. still outperforms CODEJUDGE as measured by ROC-AUC although the F1-score performance is lower in some cases.

- Computational complexity and memory cost (Reviewers FxPf, 6RBV)

We provide an estimate of computation time in Appendix F. In the base setup (attention features are computed for all layers and heads), computation of attention features is approximately two times slower than generation. However, during inference, it is possible to use only 5\% of all attention features without a significant loss of classification quality. We also highlight that we do not store all attention maps explicitly; instead, we extract the features on the fly during generation. The average memory footprint is ≈190Mb for HumanEval and ≈544Mb for MBPP. This size is a small fraction of the GPU footprint during generation.

We look forward to addressing any additional concerns and engaging in further productive discussions during the discussion phase.

Best regards,
the Authors

---

### Note · Authors · 2026-01-06

I have read and agree with the venue's withdrawal policy on behalf of myself and my co-authors.